# Postsynaptic burst reactivation of hippocampal neurons enables associative plasticity of temporally discontiguous inputs

Tanja Fuchsberger[1], Claudia Clopath[2†], Przemyslaw Jarzebowski[1†], Zuzanna Brzosko[1], Hongbing Wang[3], Ole Paulsen[1]*

[1]Department of Physiology, Development and Neuroscience, Physiological Laboratory, University of Cambridge, Cambridge, United Kingdom; [2]Department of Bioengineering, Imperial College London, London, United Kingdom; [3]Department of Physiology, Michigan State University, East Lansing, United States

**\*For correspondence:**
op210@cam.ac.uk

[†]These authors contributed equally to this work

**Competing interest:** The authors declare that no competing interests exist.

**Abstract** A fundamental unresolved problem in neuroscience is how the brain associates in memory events that are separated in time. Here, we propose that reactivation-induced synaptic plasticity can solve this problem. Previously, we reported that the reinforcement signal dopamine converts hippocampal spike timing-dependent depression into potentiation during continued synaptic activity (Brzosko et al., 2015). Here, we report that postsynaptic bursts in the presence of dopamine produce input-specific LTP in mouse hippocampal synapses 10 min after they were primed with coincident pre- and post-synaptic activity (post-before-pre pairing; Δt = –20 ms). This priming activity induces synaptic depression and sets an NMDA receptor-dependent silent eligibility trace which, through the cAMP-PKA cascade, is rapidly converted into protein synthesis-dependent synaptic potentiation, mediated by a signaling pathway distinct from that of conventional LTP. This synaptic learning rule was incorporated into a computational model, and we found that it adds specificity to reinforcement learning by controlling memory allocation and enabling both 'instructive' and 'supervised' reinforcement learning. We predicted that this mechanism would make reactivated neurons activate more strongly and carry more spatial information than non-reactivated cells, which was confirmed in freely moving mice performing a reward-based navigation task.

## Editor's evaluation

This article contains fundamental findings that substantially advance understanding of an important research question, mostly using an appropriate and validated methodology in line with the current state-of-the-art, with good and convincing support for the claims. The message of the article will have a profound and lasting influence on neuroscience.

## Introduction

For an animal to successfully adjust its behavior to changing environmental demands, it needs to learn to associate a sequence of events or actions to a subsequent outcome, for example a reward, which may be experienced minutes or hours later. How this is achieved in the brain remains unknown. Even the longest timescale synaptic plasticity event reported so far, behavioral timescale synaptic plasticity (BTSP; *Bittner et al., 2017*; *Milstein et al., 2021*), cannot bridge this temporal gap. In machine learning, reinforcement learning algorithms solve this problem by storing a temporary record of the

occurrence of an event, known as an eligibility trace, which can later undergo learning changes once the outcome is known (*Sutton and Barto, 2018*). In biology, dopamine (DA) is thought to represent such a reinforcement learning signal, converting a temporary eligibility trace into a lasting synaptic change (*Ljungberg et al., 1992*; *Schultz et al., 1997*; *Frémaux and Gerstner, 2016*; *Gerstner et al., 2018*). However, the use of a brief scalar signal to alter the relevant synaptic weights raises two fundamental problems, first, that of time scales, and second, that of specificity (or credit assignment). Whereas initial Pavlovian conditioning of reward-predicting stimuli has optimal stimulus-reward intervals of 1.5–3.0 s, other learning mechanisms occur in the second, minute, and even hour ranges (*Schultz, 2007*). Moreover, with a longer time delay, the relationship between the preceding behavior and the outcome is less clear, questioning what events or actions before the reward should be associated with the outcome. One possible mechanism that could link previous activity (e.g. exploration of an environment) with a specific outcome (e.g. finding a reward) is neuronal reactivation, or replay activity, which in the hippocampus is enhanced by reward (*Singer and Frank, 2009*; *Ambrose et al., 2016*).

The rodent hippocampus is important for spatial memories. Spatial representations are built during exploration of an environment, when the hippocampus shows theta activity, and is later reactivated or replayed both during sleep (*Wilson and McNaughton, 1994*; *Nádasdy et al., 1999*; *Lee and Wilson, 2002*) and in the awake state (*Kudrimoti et al., 1999*; *Foster and Wilson, 2006*; *Csicsvari et al., 2007*; *Diba and Buzsáki, 2007*). Reactivation occurs during sharp wave ripples (SWRs), when neurons typically fire action potentials in brief bursts (*Buzsáki et al., 1992*; *Diba and Buzsáki, 2007*). It has been reported that stimulation of dopaminergic input promotes reactivation of hippocampal cell assemblies and memory persistence (*McNamara et al., 2014*).

We investigated the effect of action potential bursts on synaptic plasticity in individual postsynaptic hippocampal CA1 neurons (SWR-associated 'reactivation') during dopaminergic modulation ('reward signal') after they had undergone a spike pairing protocol (prior exploration-based synaptic 'priming'). We found that the pairing protocol set an NMDA receptor-dependent silent eligibility trace, which could be converted several minutes later by burst activity in the presence of DA into protein synthesis-dependent long-term potentiation (LTP) mediated by a signaling pathway distinct from that of conventional LTP. Using this synaptic learning rule in a computational model we show that reactivation-induced plasticity increases specificity to reinforcement learning, offering a candidate mechanism of credit assignment in neural networks. To investigate how reactivation affects the functional properties of neurons *in vivo*, we used chronic calcium imaging of the dorsal CA1 region of the hippocampus in freely moving mice performing a reward-location learning task. We defined neuronal reactivation as activity during immobility at the reward-location in neurons that were already previously active during the reward approach. Neurons that reactivated after finding the reward had increased calcium activity and place map peaks compared to non-reactivated neurons, suggestive of changes in synaptic weights in reactivated neurons.

## Results

### Reactivation during dopaminergic modulation induces LTP

To investigate the effect of burst reactivation of individual postsynaptic CA1 cells during dopaminergic modulation, we monitored the synaptic weights of afferent synapses that had previously undergone a spike timing-dependent synaptic priming protocol. For this, we used whole-cell recording of CA1 pyramidal neurons in mouse hippocampal slices (*Figure 1A*). To be able to distinguish between conventional LTP and reactivation-induced LTP, we used a priming protocol that induces synaptic depression (*Andrade-Talavera et al., 2016*). Single postsynaptic action potentials followed by presynaptic input stimulation (post-before-pre protocol, $\Delta t = -20$ ms) led to input-specific synaptic depression in the test pathway (t-LTD; 61% ± 11% vs 100%, t(9) = 3.7, p=0.005, n=10; *Figure 1B and F*). Application of DA alone, without resuming synaptic stimulation after this pairing protocol, did not affect the depression (53% ± 7% vs 100%, t(5) = 7.0, p=0.0009, n=6; *Figure 1C and F* blue trace). Strikingly, postsynaptic action potential bursts (5–6 APs) in the presence of DA, 10 min after the pairing protocol, triggered an immediate induction of synaptic potentiation (135% ± 14.9% vs 100%, t(8) = 2.4, p=0.044, n=9; *Figure 1D and F* red trace). Burst stimulation alone, in the absence of DA, did not prevent synaptic depression (72% ± 12% vs 100%, t(6) = 2.5, p=0.048, n=7; *Figure 1E and F*

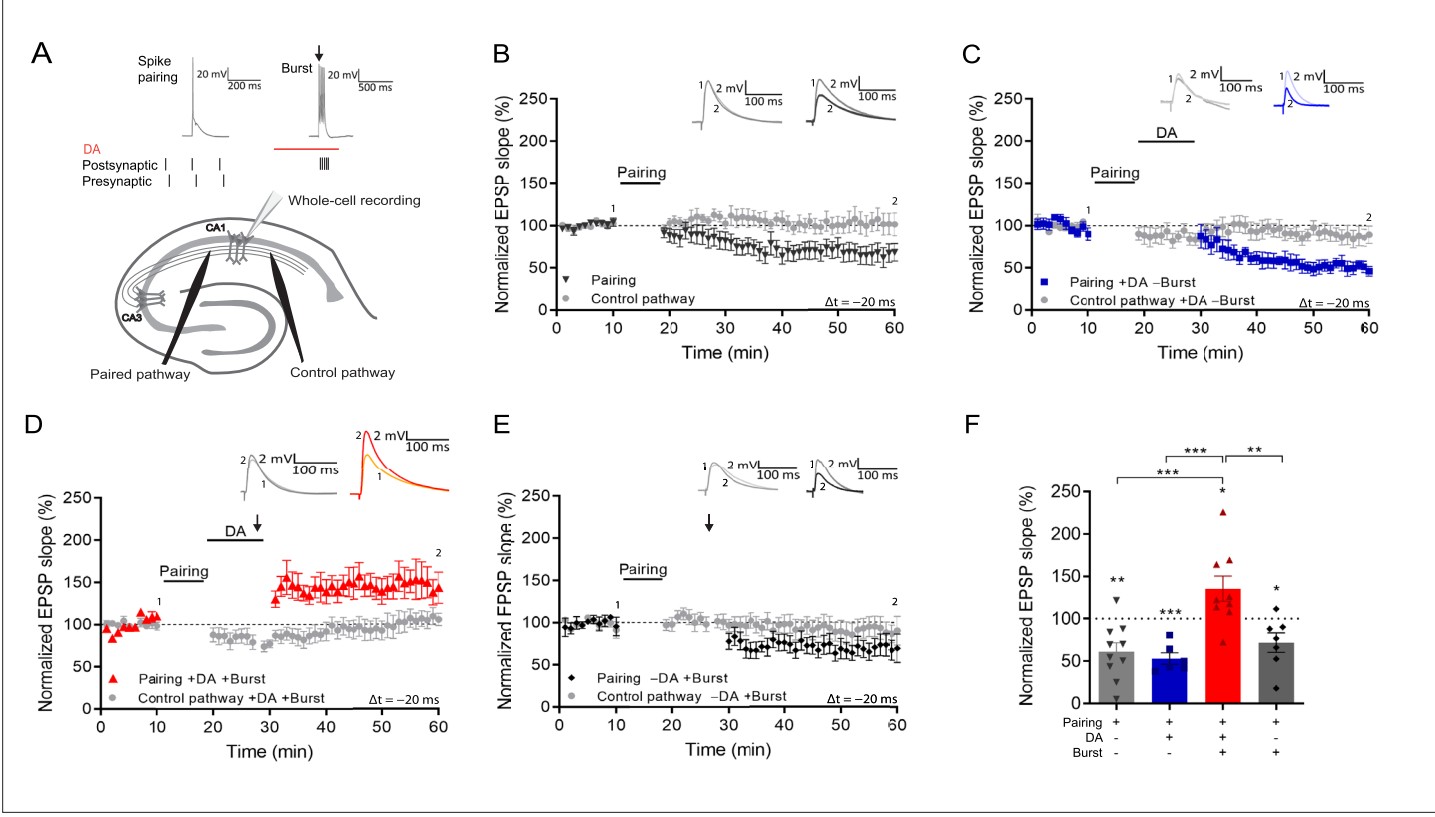

**Figure 1.** Postsynaptic burst reactivation induces LTP in the presence of dopamine (DA). (**A**), Schematic of the experimental paradigm (top) and setup (bottom).↓=6 action potential bursts; Whole-cell recording in CA1 stratum pyramidale; electrical stimulation electrodes in stratum radiatum. Plasticity was induced in one pathway (Paired pathway), and a second pathway was used for stability control and for confirmation of input specificity (Control pathway). Normalized EPSP slopes were averaged and plotted as a function of time. (**B**), Post-before pre-pairing protocol leads to input-specific synaptic depression. Pairing protocol (Δt = −20 ms) induces t-LTD (black trace) and does not affect synaptic weights in control pathway (gray trace). (**C**), DA application after a post-before-pre pairing protocol (Δt = −20 ms) does not prevent t-LTD (+DA −Burst, blue trace) and does not affect synaptic weights in control pathway (gray trace). (**D**), Application of DA together with action potential bursts of the postsynaptic cell (indicated by black arrow) induces synaptic potentiation after a post-before-pre pairing protocol (Δt = −20 ms) (+DA +Burst, red trace) and does not affect synaptic weights in control pathway (gray trace). (**E**), The same protocol, without application of DA, leads to synaptic depression (−DA +Burst, black trace) and does not affect synaptic weights in control pathway (gray trace). (**F**), Summary of results. All traces show an EPSP before (1) and 40 min after pairing (2). Plots show averages of normalized EPSP slopes ± SEM.

The online version of this article includes the following source data for figure 1:

**Source data 1.** Normalized EPSP slopes of all recorded cells.

black trace). This result suggests that the pairing protocol sets an eligibility trace allowing activated synapses to be selectively altered minutes later by reactivation of the postsynaptic neuron in the presence of DA. To our knowledge, these are the longest-lasting synaptic eligibility traces reported in the brain.

We next investigated the requirements for setting the eligibility trace. First, we found that synaptic potentiation was indeed not observed without prior spike pairing (93% ± 8% vs 100%, t(6) = 0.84, p=0.43, n=7; *Figure 2A*). Induction of hippocampal t-LTD requires metabotropic glutamate receptors (mGluRs; *Andrade-Talavera et al., 2016*). To investigate whether LTD or mGluR signaling is required for burst-induced potentiation, we applied the mGluR antagonist MPEP throughout the recording. This blocked t-LTD (94% ± 6% vs 100%, t(6) = 0.95, p=0.38, n=7; *Figure 2B and C*, blue; vs control t-LTD 69% ± 7%, n=7; t(11.5) = 2.67, p=0.02; *Figure 2B and C*, black) but burst-induced potentiation was intact (131% ± 10% vs 100%, t(6) = 3.07, p=0.021, n=7; *Figure 2B and C*, red), suggesting that setting the eligibility trace by spike pairing is distinct from the signaling mechanism that mediates t-LTD.

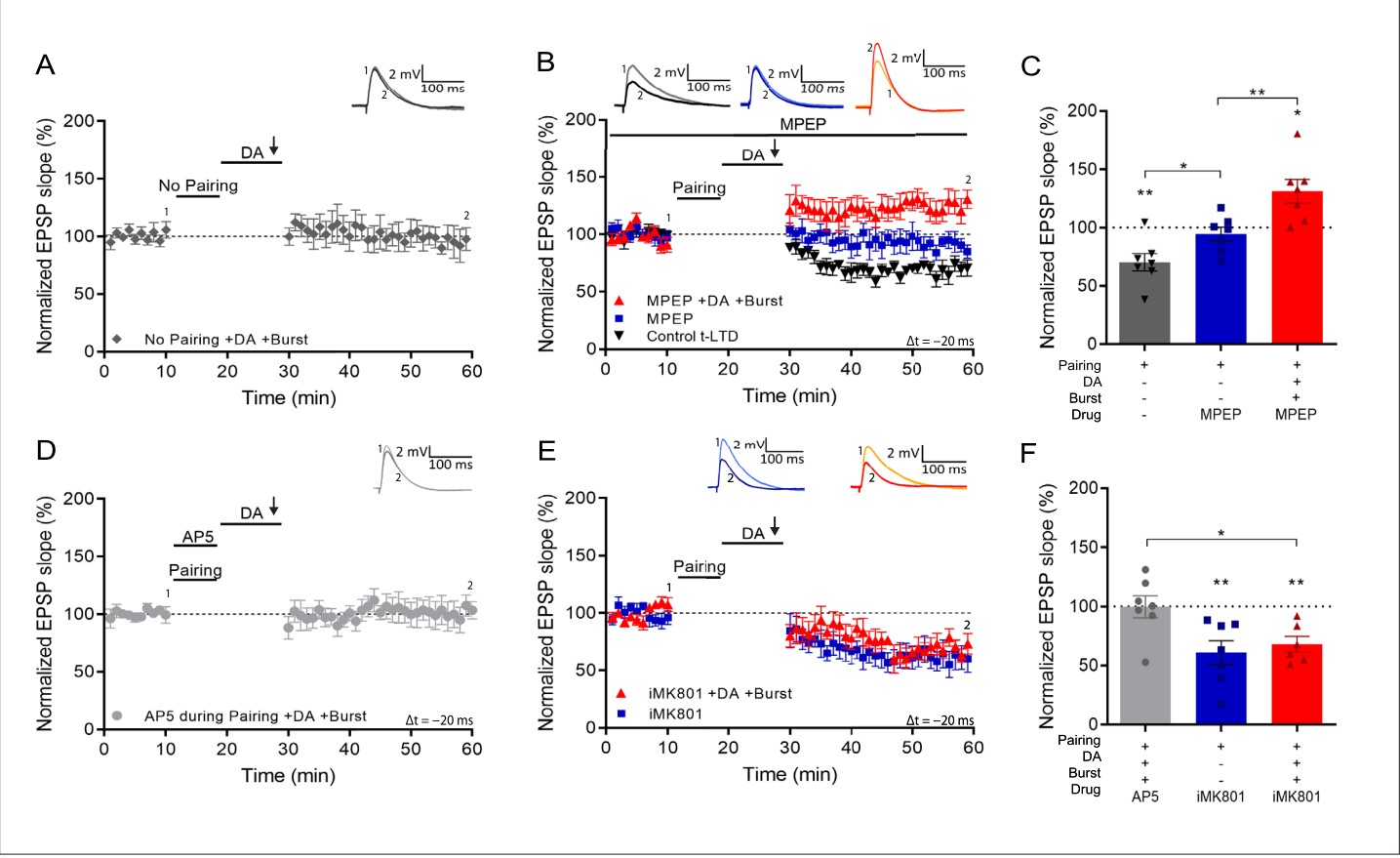

**Figure 2.** Setting of the eligibility trace is independent of LTD but requires postsynaptic NMDARs. (**A**), Application of DA and burst stimulation, but without pairing protocol, does not induce potentiation. (**B**), Post-before-pre pairing protocol (Δt = –20 ms) leads to synaptic depression (t-LTD) (black trace), which is blocked by MPEP (blue trace). MPEP does not block DA- and burst-induced potentiation (red trace). (**C**), Summary of results. (**D**), Application of AP5 during pairing blocks DA- and burst-induced potentiation. (**E**), Postsynaptic intracellular MK801 (iMK801) does not block t-LTD (blue trace) but blocks DA- and burst-induced potentiation (red trace). (**F**), Summary of results. All traces show an EPSP before (1) and 40 min after pairing (2). Plots show averages of normalized EPSP slopes ± SEM.

The online version of this article includes the following source data for figure 2:

**Source data 1.** Normalized EPSP slopes of all recorded cells.

Many forms of hippocampal plasticity require NMDA receptors (NMDARs; *Shipton and Paulsen, 2014*). We, therefore, asked if activation of NMDARs during the pairing protocol is required for the synaptic eligibility trace to be set. We found that application of the NMDAR antagonist D-AP5 during pairing (Δt = –20 ms) abolished both t-LTD and the subsequent burst-induced potentiation, resulting in no change of synaptic weights (100% ± 9% vs 100%, t(6) = 0.043, p=0.97, n=7; *Figure 2D and F*). We then investigated whether specifically postsynaptic NMDARs are required for the eligibility trace. Loading the postsynaptic cell with the NMDAR channel blocker MK801 through the recording pipette did not affect t-LTD (68% ± 7% vs 100%, t(5) = 4.77, p=0.005, n=6; *Figure 2E and F*) but completely abolished burst-induced potentiation, leaving synaptic depression instead (61% ± 10% vs 100%, t(6) = 3.85, p=0.0084, n=6; *Figure 2E and F*). These results show a double dissociation between t-LTD induced by mGluR and non-postsynaptic NMDAR signaling and the eligibility trace for reactivation-induced potentiation set by postsynaptic ionotropic NMDARs.

We next investigated the mechanism that induces synaptic potentiation during burst stimulation. It was previously shown that activation of NMDARs after pairing is necessary to induce DA-dependent potentiation with subthreshold synaptic stimulation (*Brzosko et al., 2015*). However, the NMDAR antagonist D-AP5, applied after the pairing protocol but before burst stimulation, did not affect burst-induced potentiation (127% ± 9%, t(5) = 2.94, p=0.032, n=6; *Figure 3A and C*). In contrast, voltage-gated calcium channels (VGCCs) are required as application of nimodipine, an L-type VGCC blocker,

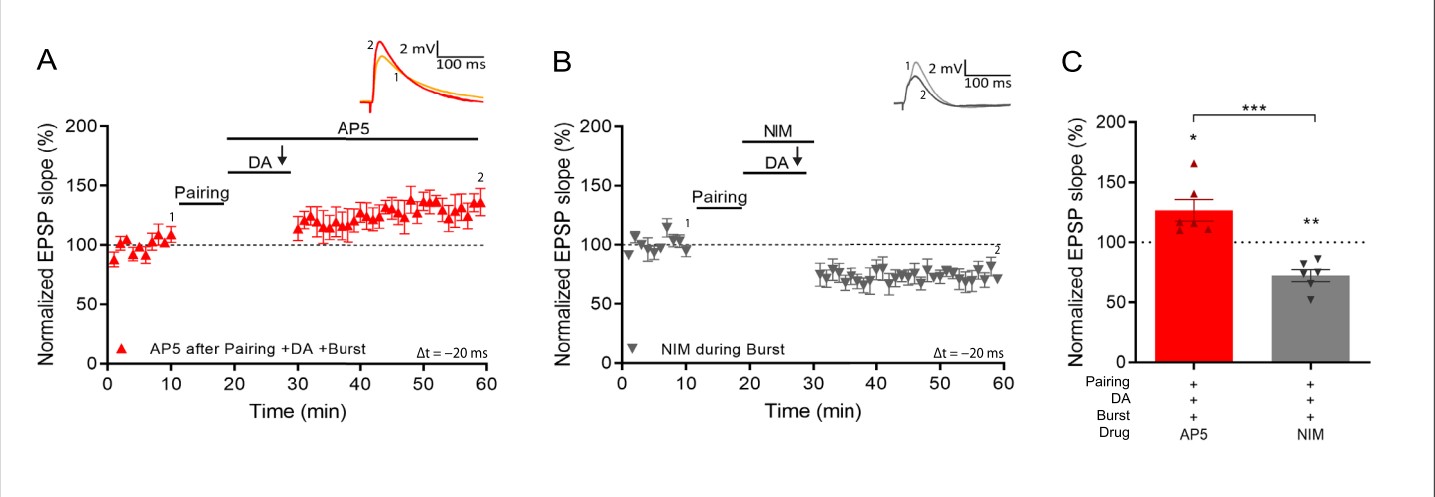

**Figure 3.** Voltage gated calcium channels mediate DA and reactivation-induced plasticity during burst stimulation. Normalized EPSP slopes were averaged and plotted as a function of time. (**A**), Application of AP5 after pairing does not prevent DA- and burst-induced potentiation. (**B**), Application of nimodipine during the burst prevents DA- and burst-induced potentiation leaving synaptic depression. (**C**), Summary of results. All traces show an EPSP before (1) and 40 min after pairing (2). Plots show averages of normalized EPSP slopes ± SEM.

The online version of this article includes the following source data for figure 3:

**Source data 1.** Normalized EPSP slopes of all recorded cells.

before the burst completely abolished LTP and left synaptic depression instead (72% ± 5% vs 100%, t(5) = 5.6, p=0.0026, n=6; *Figure 3B and C*) indicating that, as in some other forms of synaptic potentiation (*Grover and Teyler, 1990*), Ca$^{2+}$ entry through VGCCs is required for burst reactivation to induce LTP.

## Signaling pathway mediating reactivation-induced LTP

These findings suggest that coincident DA signaling and postsynaptic Ca$^{2+}$ increase enable the potentiation of previously primed synapses. Searching for a potential coincidence detector for DA and intracellular Ca$^{2+}$, we focused on adenylyl cyclases (ACs). They are activated by Gs-coupled dopamine D1/D5 receptor stimulation (*Neve et al., 2004*), and subtypes AC1 and AC8 are additionally Ca$^{2+}$-stimulated (*Wayman et al., 1994*; *Watson et al., 2000*; *Ferguson and Storm, 2004*). To investigate whether AC subtypes AC1 and/or AC8 are involved in the form of plasticity described here, we tested the induction protocol in AC1/AC8 double knockout (AC DKO) mice (*Wong et al., 1999*). When postsynaptic burst stimulation in the presence of DA was applied after a negative pairing protocol (Δt = –20 ms; *Figure 4Ai*) in slices from AC DKO mice, the conversion to potentiation was absent and significantly different from DA- and burst-induced potentiation in slices from wildtype mice (AC DKO, 90% ± 8%, n=6 vs WT, 132% ± 11%, n=8; t(12) = 2.8, p=0.015; *Figure 4Bi and Biii*), revealing a role for AC1/AC8 as coincidence detector for DA- and Ca$^{2+}$-induced potentiation. In contrast, conventional, DA-independent t-LTP induced by a pre-before-post pairing protocol (Δt = +10 ms; *Figure 4Aii*) showed significant potentiation comparable to that seen in wildtype mice (AC DKO, 150% ± 19% vs 100%, t(5) = 2.6, p=0.049, n=6; WT, 163% ± 14% vs 100%, t(9) = 4.4, p=0.0015, n=10; *Figure 4Bii and Biii*).

AC activation produces an increase in cyclic adenosine monophosphate (cAMP) which activates protein kinase A (PKA; *Sassone-Corsi, 2012*). To test whether this signaling cascade is required for reactivation-induced potentiation, we loaded the postsynaptic cell with a PKA blocker, protein kinase inhibitor-(6-22)-amide, through the recording pipette. In this configuration burst stimulation in the presence of DA after the post-before-pre protocol failed to induce synaptic potentiation (69% ± 13% vs 100%, t(5) = 2.4, p=0.064, n=6; *Figure 4Ci and Ciii*). In contrast, conventional pre-before-post pairing induced significant potentiation, albeit of somewhat reduced magnitude (137% ± 14% vs 100%, t(5) = 2.63, p=0.0463, n=6; *Figure 4Cii and Ciii*).

The requirement of DA and PKA for burst-induced potentiation is shared with late-phase LTP (*Frey et al., 1990*; *Frey et al., 1993*), which requires protein synthesis (*Frey et al., 1988*). We therefore

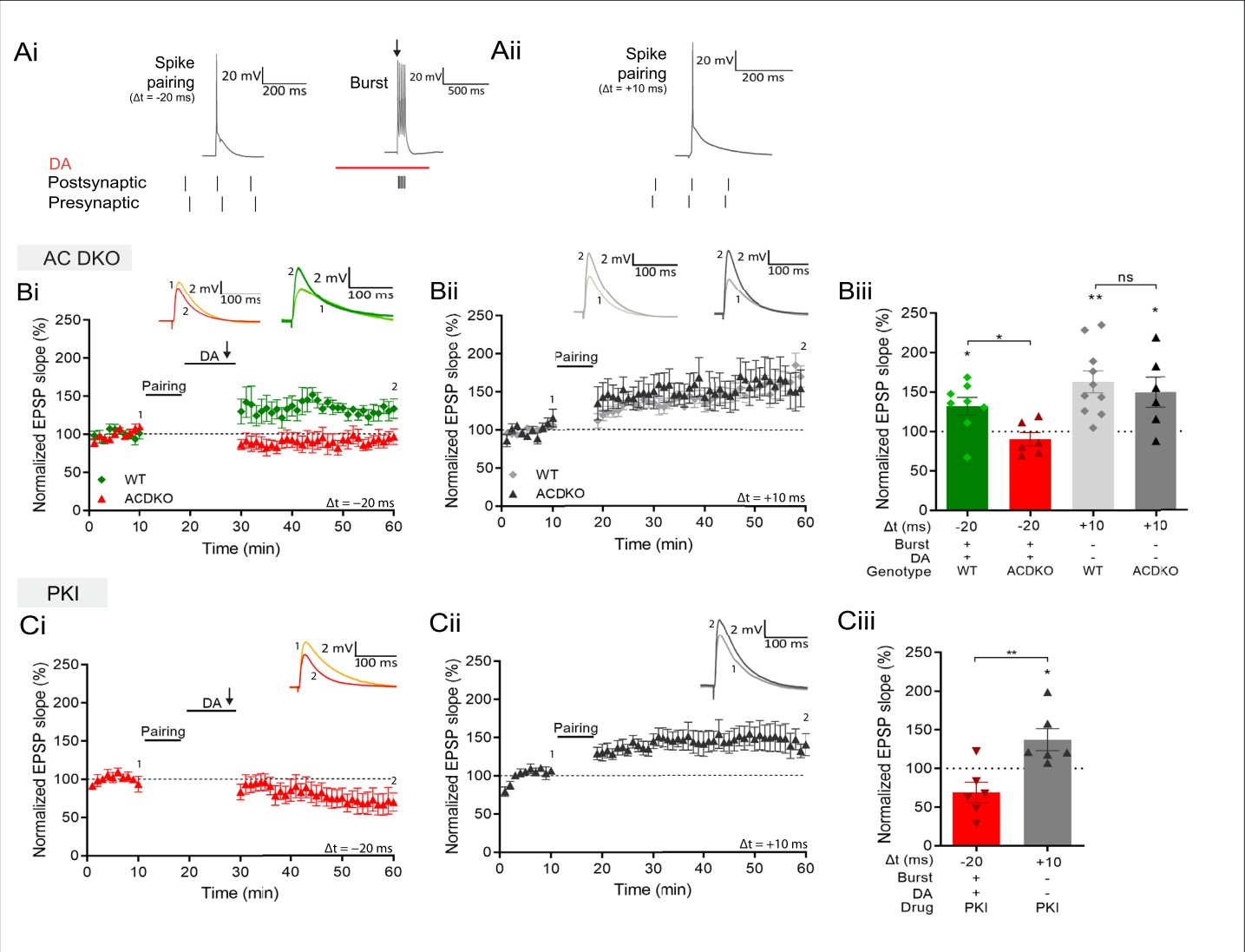

**Figure 4.** DA and reactivation-induced potentiation require AC1/AC8 and PKA. Schematics show the difference between the three induction protocols. Normalized EPSP slopes were averaged and plotted as a function of time. (**Ai**), DA and burst stimulation after a post-before-pre pairing protocol (Δt = –20 ms) induces potentiation. (**Aii**), A pre-before-post pairing protocol induces t-LTP (Δt = +10 ms). (**B**), AC DKO mice do not show DA-dependent plasticity with postsynaptic bursts (**Bi**) but shows conventional t-LTP (**Bii**). Summary of results (**Biii**). (**C**), Postsynaptic application of protein kinase inhibitor-(6-22)-amide (PKI) prevents DA-dependent plasticity with postsynaptic bursts (**Ci**) but leaves conventional t-LTP intact (**Cii**). Summary of results (**Ciii**). All traces show an EPSP before (1) and 40 minutes after pairing (2). Plots show averages of normalized EPSP slopes ± SEM.

The online version of this article includes the following source data for figure 4:

**Source data 1.** Normalized EPSP slopes of all recorded cells.

investigated whether the burst-induced rapid potentiation also requires protein synthesis by delivering the protein synthesis inhibitor anisomycin (AM) to the postsynaptic cell through the recording pipette. We found that, with anisomycin, burst stimulation in the presence of DA no longer induced conversion to potentiation but left a synaptic depression instead, which was significantly different from vehicle control (AM, 52% ± 11% vs 100%, t(5) = 4.5, p=0.0062, n=6; *Figure 5Ai and Aiii*, red; vs vehicle, 128% ± 17%, n=5; t(9) = 3.9, p=0.0033; *Figure 5Ai and Aiii*, green). In contrast, conventional t-LTP induced by pre-before-post pairing was unaffected by anisomycin (161% ± 20% vs 100%, t(5) = 3.0, p=0.029, n=6; *Figure 5Aii and Aiii*, black). Furthermore, the post-before-pre pairing protocol induced t-LTD under these conditions (65% ± 6% vs 100%, t(5) = 5.9, p=0.0019, n=6; *Figure 5Aii and Aiii*, gray). We confirmed that postsynaptically applied anisomycin did not affect synaptic responses in baseline conditions (95% ± 9.7% vs 100%, t(6) = 0.56, p=0.59, n=7; *Figure 5—figure supplement*

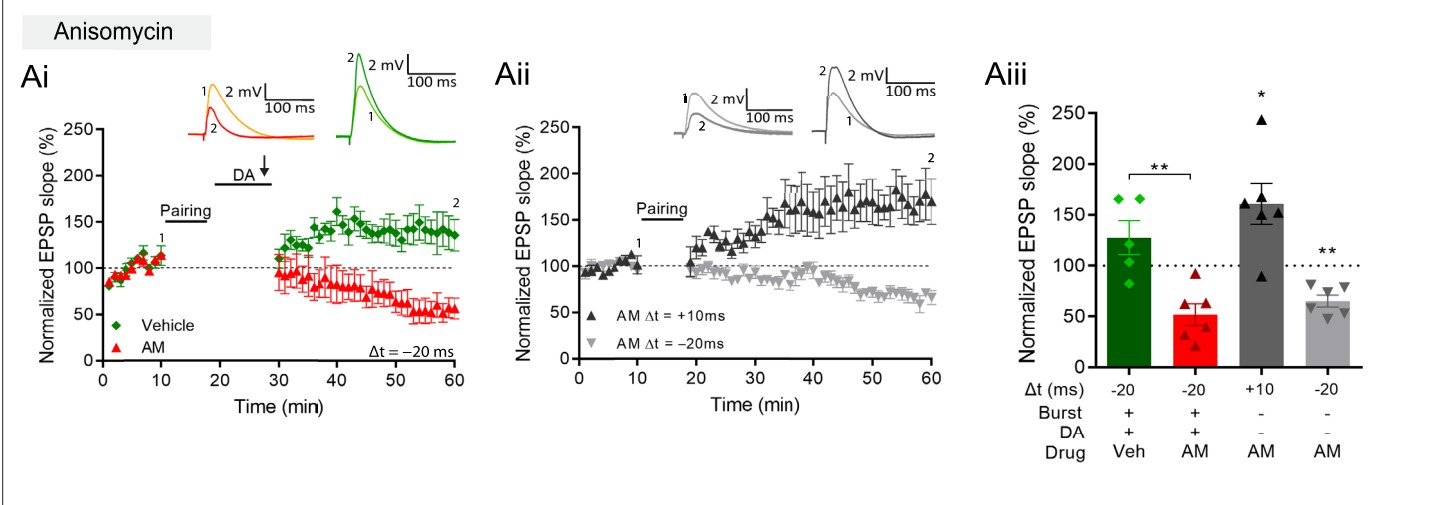

**Figure 5.** DA and reactivation-induced plasticity requires protein synthesis. (**A**), Postsynaptic anisomycin prevents DA-dependent plasticity with postsynaptic burst stimulation (**Ai**), but leaves conventional t-LTD (**Aii**, gray trace) and t-LTP (black trace) intact. Summary of results (**Aiii**). All traces show an EPSP before (1) and 40 min after (2) pairing. Plots show averages of normalized EPSP slopes ± SEM.

The online version of this article includes the following source data and figure supplement(s) for figure 5:

**Source data 1.** Normalized EPSP slopes, spike amplitudes and spike width of all recorded cells.

**Figure supplement 1.** Postsynaptic anisomycin (AM) does not affect baseline synaptic responses and action potential properties.

**1A**). Furthermore, we compared action potential properties during pairing in cells with anisomycin to cells loaded with vehicle controls. Spike amplitude (AM 112 mV ± 3 mV, Vehicle 111 mV ± 3 mV) and spike width (AM 3.3 ms ± 0.2 ms, Vehicle 3.3 ms ± 0.2 ms) showed no significant differences (amplitude t(10) = 0.09050, p=0.92; width t(10) = 0.1134, p=0.91; *Figure 5—figure supplement 1B, C*).

## Burst-dependent plasticity increases specificity in reinforcement learning models

These experimental results show that, after a priming event, burst reactivation in the presence of DA induces a rapid form of protein synthesis-dependent LTP. This mechanism would ensure that only salient neuronal activity induces long-term changes in the network. We implemented this synaptic learning rule in a computational model to explore how such plasticity would control learning in a feedforward artificial neural network resembling hippocampal CA1. The learning rule states that the change in synaptic weights $\Delta w$ between input and output neurons ($inp \rightarrow o$) depends on an eligibility trace $e$ (set during the initial priming event), the reinforcement signal (dopamine $d$), and reactivation (bursting activity $b$).

$$\Delta w_{inp \rightarrow o} = \alpha_{LTP} e d b \tag{1}$$

The parameter $\alpha_{LTP}$ is the learning rate. When there is no DA or bursts during the trial, the rule is updated to result in depression proportional to the eligibility trace with a learning rate $\alpha_{LTD}$ (see Methods). When using a standard reinforcement learning (RL) rule, which does not depend on burst reactivation, all previously primed synapses are potentiated after receiving the DA signal (*Figure 6A*). Thus, the global neuromodulatory signal in traditional RL models provides limited information. In contrast, when applying the burst-dependent learning rule (*Equation 1*) to the network, in which potentiation of primed synapses depends on both the reward signal and reactivation, a selected subset of inputs becomes potentiated, while inputs on non-reactivated neurons remain depressed (gray) (*Figure 6B*). During DA modulation, information is allocated to primed synapses by reactivation of the postsynaptic neuron, and the broader computational implications of this learning rule depend on the control of postsynaptic neuronal bursting activity. First, it is possible that neurons are recruited to a new memory trace based on their relative neuronal excitability before the training session as suggested by the memory allocation hypothesis (*Yiu et al., 2014*). According to this scenario, the

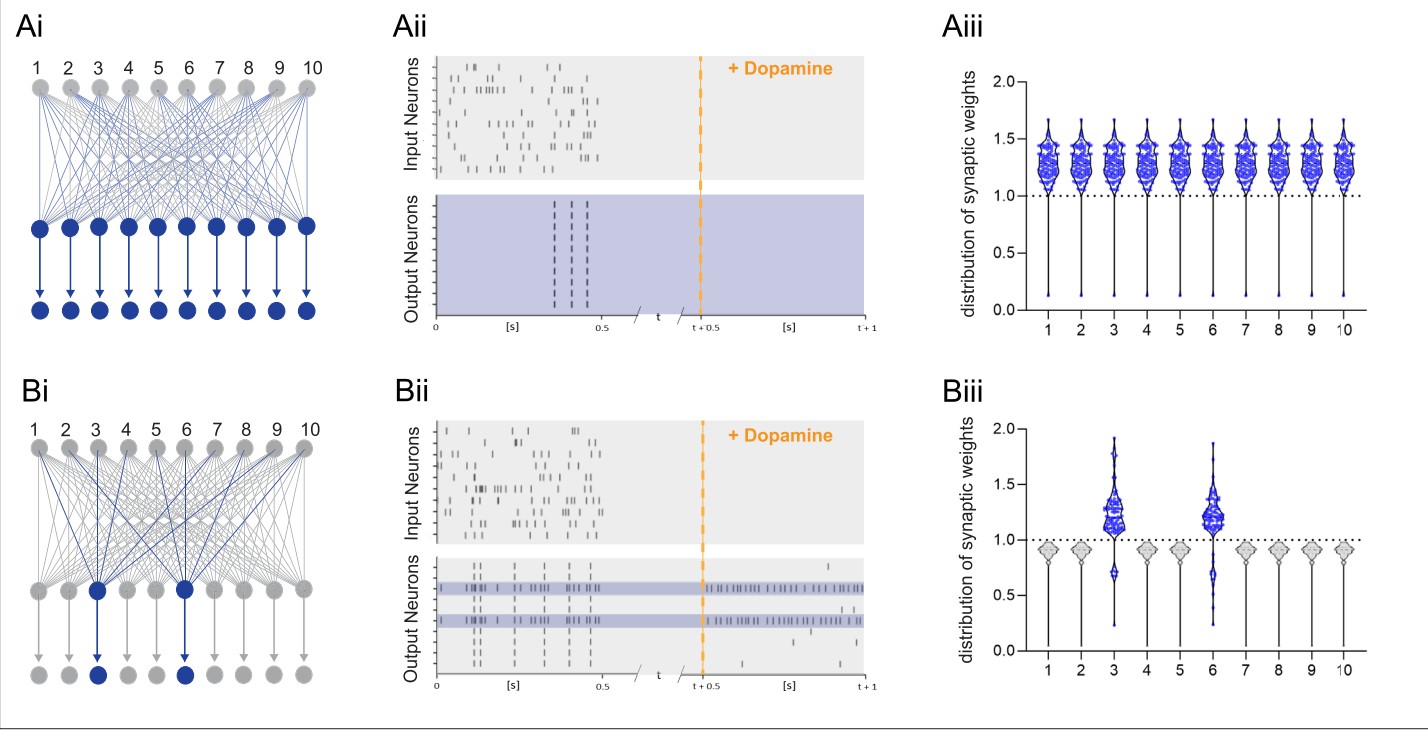

**Figure 6.** Dopamine-dependent burst-induced plasticity rule reduces the number of neurons in the network encoding a memory. (**Ai-iii**), Standard reinforcement learning rule shows reward associating inputs 1–10 (blue) with potentiation of all synaptic weights (blue). (**Bi-iii**), Burst-dependent potentiation reduces the number of neurons encoding the memory, leading to potentiation of synapses exclusively onto the most excitable burst-firing neurons 3, 6 (blue).

The online version of this article includes the following source data for figure 6:

**Source data 1.** Raster plot data and synaptic weights.

most excitable cells would be the most likely to show action potential bursts during reactivation and, therefore, show synaptic potentiation. Alternatively, there is evidence for replay of prioritized experience (*Igata et al., 2021*), suggesting that cells encoding the most salient events preceding the reward would reactivate, and thereby determine which set of cells would show potentiation during reward (*Figure 6B*). Assuming prioritized experience reflects experience relevant to the reward, this could help credit assignment in the network. In addition, because of the exclusive requirement of postsynaptic activity for potentiation to occur, this mechanism offers another intriguing possibility, namely that other inputs active at the reward location carries additional information about the nature of the reward, e.g., food, or the reward location, such as the presence of specific landmarks, which could elicit postsynaptic bursting activity. Under this hypothesis, neuronal reactivation would serve to associate a specific outcome to the priming event. When different instructive inputs induce bursting each in distinct subsets of neurons during reward, selective increases in synaptic weights would not only enable the encoding of reward but also distinguish between different rewards (*Figure 7A*). Finally, we explore how the learning rule performs in a network supervised by feedback synaptic input to strengthen synapses onto specific neurons encoding reward-related features. By allowing feedback input to assign which part of the network is responding, the burst-dependent learning rule enables the network to selectively learn relevant information (*Figure 7B*), resulting in potentiation of those synapses (magenta and cyan in *Figure 7Biii*) that are active temporally separated, but less when simultaneous active (*Figure 7Biv*). This provides a mechanism to associate temporally separated, reward-relevant information in a neuronal network. The burst-dependent learning rule provides the network with a gating mechanism for memory allocation to an engram. Thus, burst-induced plasticity reduces the number of neurons encoding the reward location and other reward-related information, increasing the specificity of synaptic memory in a neuronal network.

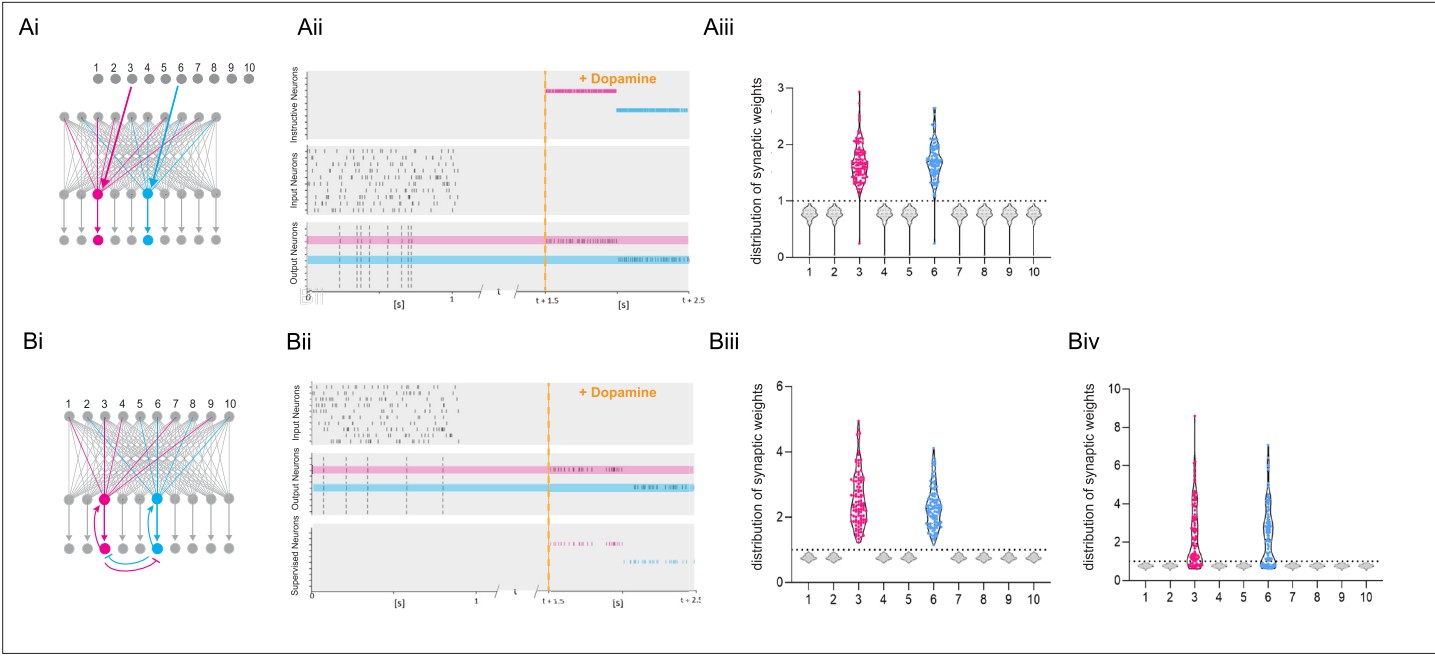

**Figure 7.** Dopamine-dependent burst-induced plasticity rule enables reinforcement learning (RL) models to encode a specific salient event. (**Ai-iii**), Instructive RL rule allows two inputs that code for different information to store the memory in separate sets of neurons, thus encoding not only the reward, but also other reward-relevant features 3, 6 (magenta, cyan). (**Bi-ii**) A supervised network enables burst-eliciting feedback synaptic input to assign credit to select synapses in the network to encode a desired reward identity. (**Biii**) Time-dependent lateral inhibition at the output neurons suppress non-relevant information. When only one of the inputs is active, the animal can learn two different memories over time in neurons 3, 6 (magenta, cyan). (**Biv**) When both inputs are active at the same time they compete with each other, and synapses onto these neurons (magenta, cyan) are less potentiated.

The online version of this article includes the following source data for figure 7:

**Source data 1.** Raster plot data and synaptic weights.

### Increased calcium responses and spatial information in reactivated CA1 place cells

Based on these results we predicted that, when an animal navigates toward a reward, the previously reactivated hippocampal neurons would be more strongly activated than non-reactivated neurons. To test this prediction, we monitored calcium transients in hippocampal excitatory cells with a head-mounted microscope (*Ghosh et al., 2011*) while mice navigated on a 'cheeseboard' maze (*Dupret et al., 2010*) with two reward locations, one new to the animal and one previously learnt (*Figure 8A*). We defined neuronal reactivation as activity during immobility in neurons previously active during locomotion. Cells that were active when mice moved towards the reward locations were classified as reactivated at reward if they were active again during immobility when mice consumed reward (*Foster and Wilson, 2006*; *O'Neill et al., 2006*; *Csicsvari et al., 2007*) and non-reactivated if no further calcium event was detected after they had reached the reward. Of the cells that were active on the maze in a given trial, 44 ± 1% were reactivated at either or both of the reward locations. There was no detectable difference between the number of cells reactivating at the previously learnt or new reward location (*Figure 8B*). Less frequent were the reactivations during immobility at non-rewarded locations where 15 ± 2% of cells reactivated (*Figure 8B*).

To compare the strength of neuronal activation, we measured area-under-curve (AUC) of calcium events occurring before and after the reactivation (*Figure 8C*). Cells with the largest activity peaks during locomotion were the most likely to reactivate at the reward location. Following their first reactivation, they had larger activity peaks than the previously non-reactivated cells (0.54 ± 0.01 vs 0.42 ± 0.01 of the cell's max AUC, $F_{(1,16)}$ = 10.12, p=0.006, *Figure 8D*). The effect on activity peaks was specific to reward locations, and reactivation at other locations did not affect activity peaks in the following trials ($F_{(1, 16)}$=0.56, p=0.47, *Figure 8D*). The effect of reactivation was independent of the cell event rates: of two cell groups with matching event rates in a given trial, the one whose cells were

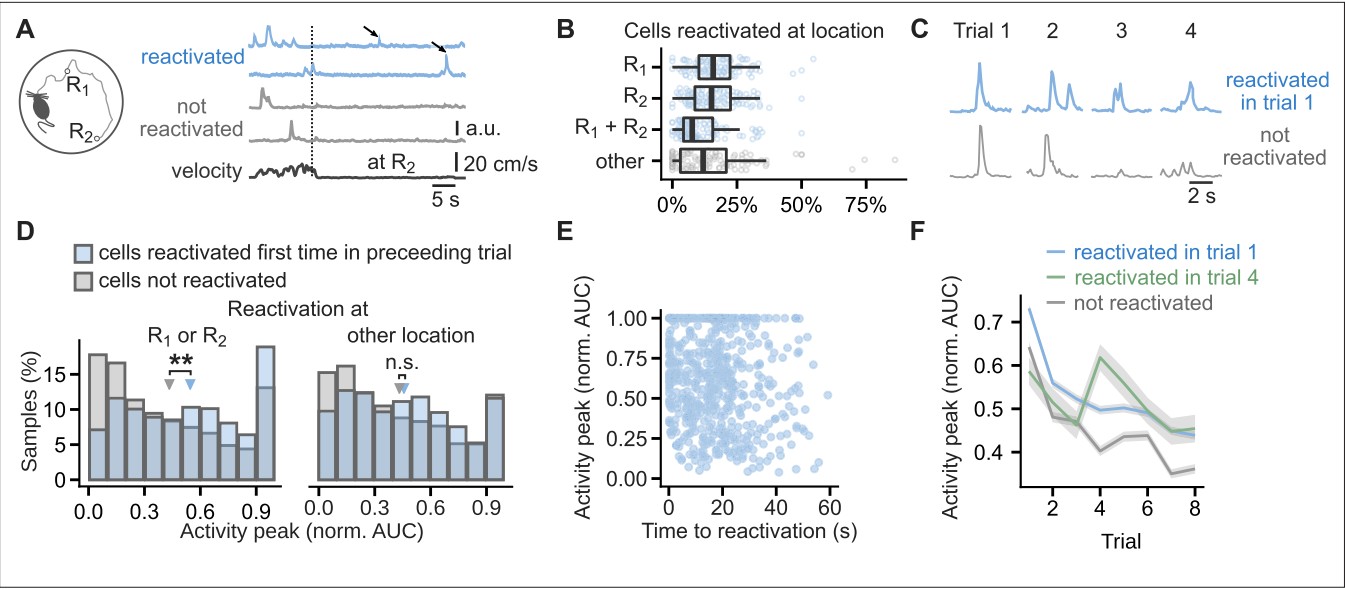

**Figure 8.** Cells that reactivated at reward location have higher activity peaks than non-reactivated cells. (**A**), Running path of a mouse (left) and calcium traces of example cells (right) in a representative single trial. Mice ran toward two reward locations: one previously learned (**$R_1$**) and one new (**$R_2$**). Calcium traces for two cells that reactivated at the reward location and two that did not. Arrows mark reactivations. Vertical line marks time of arrival at $R_2$. (**B**), Percentage of cells that reactivated at one or both of the reward locations or other locations. Points mark percentage in single trials. Box-and-whisker plots: median, 25 and 75th percentile, and extreme values. n=444 trials. (**C**), Traces centered on activity peaks for two example cells: one that reactivated at the new reward location in trial 1 (top) and one that did not reactivate in any of the trials (bottom). (**D**), Histogram of activity peaks normalized to maximum AUC value compared between cells that reactivated for the first time in the preceding trial and cells that did not reactivate in any of the trials. Triangles mark mean values. Permutation tests for repeated measures ANOVA, model for reactivations at the reward locations: significant effect of trial ($F_{(6, 96)}$=4.51, p<0.001) and reactivation ($F_{(1,16)}$ = 10.15, p=0.006), model for reactivations at non-rewarded locations: significant effect of trial ($F_{(6, 96)}$=7.85, p=0.001) and non-significant of reactivation ($F_{(1, 16)}$=0.56, p=0.47). n=6785 cell trials from 1405 cells. (**E**), The duration from activity during locomotion to the time of first reactivation did not correlate with activity peaks in the following trial, suggesting a long-lasting eligibility trace.(**F**), Mean calcium peak in each trial for cells that reactivated in trial 1 or trial 4 compared to cells that did not reactivate in any trial. Ribbons extend +/- 1 SEM. Cells that reactivated in trial 1 had significantly higher normalized calcium peaks in all trials. Cells that reactivated for the first time in trial 4 had significantly higher normalized calcium peaks in trials 4, 5, 7 and 8 but not in trial 6 and trials before the reactivation. Permutation t-tests with Benjamini-Hochberg correction for multiple comparisons and between-animal random effects. n=6721 cell trials from 681 reactivated in trial 1 cells, n=769 from 96 reactivated in trial 4 cells, and n=5980 from non-reactivated 607 cells. **p<0.01.

The online version of this article includes the following figure supplement(s) for figure 8:

**Figure supplement 1.** Similar effect of the reactivation on the activity peaks in different cell groups.

**Figure supplement 2.** Place cells during learning in reactivated and non-reactivated neurons.

reactivated at reward for the first time, had larger peaks in the following trial ($F_{(1, 16)}$=7.54, p=0.02, *Figure 8—figure supplement 1A*). The effect of reactivation at reward had two other similarities with the synaptic plasticity experiments: a long-lasting eligibility trace and persistence of the change. To test the first, we confirmed that the activity peaks in the trial following the reactivation did not correlate with the time from the activity during locomotion to the reactivation (*Figure 8E*). To test the persistence of the change, we confirmed that the reactivated cells maintained higher calcium activity peaks than non-reactivated cells throughout the later trials (*Figure 8F*, *Figure 8—figure supplement 1B*). Typically, the peaks during locomotion increased in the trial immediately before the first reactivation at reward (*Figure 8—figure supplement 1B*), suggesting that more excited cells are the ones that undergo the change. The calcium activity peaks we report could be affected by photobleaching of the GCaMP6f sensor. As the reactivated cells were more active than non-reactivated cells (*Figure 8—figure supplement 1A*), we would expect them to be more affected by photobleaching, but nevertheless, the magnitude of their calcium transients remained higher than in the non-reactivated cells.

60% of the reactivated cells (n=971 from 7 mice) and 42% of the non-reactivated cells (n=284 from 7 mice) showed activity at specific locations and were classified as place cells (see Methods). Both place cells and non-place cells showed higher calcium peaks following reactivation and the increase in place cells was not significantly different from that in non-place cells ($F_{(1,16)}$ = 0.11, p=0.75,

*Figure 8—figure supplement 1C, D*). To investigate any learning-induced changes in place maps we assessed how the location-averaged activity in place cells changed with reactivation from the first to the second half of the trials (change from trials 1–4 to trials 5–8, *Figure 8—figure supplement 2A*). The place map peaks increased significantly more in the place cells that reactivated in trial 4 or later than in the non-reactivated place cells (*Figure 8—figure supplement 2B*, place map peak ratio 1.4 ± 0.1 vs 1.1 ± 0.1, n=194 reactivated and n=306 non-reactivated place cells; permutation test for repeated measures ANOVA: $F_{(1, 16)}$=11.3, p=0.008). The change in spatial information also significantly differed between the two groups (*Figure 8—figure supplement 2C*, change by 0.011 ± 0.016 to −0.047 ± 0.014 a.u.; permutation test for repeated measures ANOVA: $F_{(1, 16)}$=4.4, p=0.048), but there was no significant difference in place map stability (*Figure 8—figure supplement 2E*, correlation of 0.45 ± 0.02 vs 0.47 ± 0.02; permutation test for repeated measures ANOVA: $F_{(1, 16)}$=0.29, p=0.59). The time from the activity during locomotion to the reactivation did not correlate with a change in spatial information (*Figure 8—figure supplement 2D*). Reactivated place cells conveyed more spatial information in late trials (trials 5–8) compared to place cells that did not reactivate (0.15 ± 0.01 vs 0.12 ± 0.01 a.u.; permutation test for repeated measures ANOVA: $F_{(1, 16)}$=4.4, p=0.055). This learning-associated increase in calcium response and spatial information supports a reactivation-dependent LTP-like mechanism *in vivo* (*Cacucci et al., 2007*).

## Discussion

In summary, we investigated the effects of postsynaptic neuronal reactivation on hippocampal synaptic plasticity, reinforcement learning, and spatial coding. We found that postsynaptic burst reactivation of CA1 pyramidal cells in the presence of the reward signal DA rapidly potentiates synapses that have previously undergone a spike timing-dependent priming protocol, providing direct evidence for reactivation-induced synaptic plasticity. A computational model showed how this learning rule increases specificity in reinforcement learning models. Recordings from freely moving mice showed that neurons that reactivated at reward locations had enhanced CA1 place cell calcium signals and carried more spatial information than cells that did not reactivate.

The results suggest that reactivation-induced plasticity is mediated by two sequential coincidence detectors: postsynaptic NMDARs detecting coincident pre- and postsynaptic activity and AC1/AC8 as coincidence detector of DA and reactivation-induced $Ca^{2+}$ increase. Although we used a spike pairing protocol, we cannot exclude the possibility that activation of postsynaptic NMDA receptors without postsynaptic action potentials would be sufficient to set the eligibility trace.

AC1/AC8 is synergistically activated when the two signals, Gs-coupled dopamine D1/D5 receptor activation and $Ca^{2+}$ influx, occur at the same time (*Wayman et al., 1994*; *Watson et al., 2000*; *Ferguson and Storm, 2004*; *Neve et al., 2004*). The time course of DA signaling depends on brain area, firing mode (tonic or phasic) of dopaminergic cells, DA release and diffusion as well as time course of intracellular signaling pathways (*Liu et al., 2021*). Recent developments of fluorescent DA sensors (*Sun et al., 2018*; *Sun et al., 2020*; *Elizarova et al., 2022*) would enable monitoring the precise time course of DA in the hippocampus in future studies. Moreover, to investigate the precise timing requirements for DA-dependent reactivation-induced plasticity further, uncaging of caged DA or optogenetically-induced DA release would be suitable approaches for temporal control of the DA transient.

Our experiments were carried out at 2 mM external calcium concentration, which is a standard calcium concentration used in most *ex vivo* plasticity experiments, but above the reported ionic calcium concentration in rat and human cerebrospinal fluid (*Jones and Keep, 1988*; *Forsberg et al., 2019*). Unfortunately, the extracellular calcium concentration at synaptic sites is not known, as discussed elsewhere (*Lopes and Cunha, 2019*). It was recently reported that burst pairing, but not pairings of single pre- and postsynaptic action potentials, induces synaptic plasticity at 1.3 mM extracellular calcium in rat hippocampal slices (*Inglebert et al., 2020*). It will be interesting to investigate whether, even if single-spike pairing might not induce plasticity at low calcium concentrations, it would still be sufficient for the initial priming of those synapses. If that were the case, this would set the conditions to enable reactivation-induced plasticity, which relies on bursts, and would hold also in low-calcium conditions. The signaling cascade leading to synaptic potentiation involves postsynaptic PKA and protein synthesis, which are not required for conventional early LTP (*Park et al., 2014*). Traditionally, LTP has been classified into early- and late-phase LTP (*Frey et al., 1988*; *Frey et al., 1990*), and it has

been reported that dopaminergic signaling is required for maintenance of late-phase LTP (*Frey et al., 1990*; *Huang and Kandel, 1995*; *Matthies et al., 1997*). The DA-dependent form of plasticity we describe here shares properties with 'late-phase' LTP, including a role of postsynaptic action potentials during induction (*Dudek and Fields, 2002*) and a requirement of protein synthesis for expression (*Frey et al., 1988*; *Huang and Kandel, 1995*). However, it is remarkably fast, suggesting a dissociation between different signaling pathways, rather than different temporal phases of LTP. The mechanism we describe here would be compatible with some of the key concepts of the 'revised' synaptic tagging hypothesis (*Redondo and Morris, 2011*). Specifically, our findings strongly support the view that the fate of a memory is not determined at the time of encoding. This is based on the finding that plasticity induction can lead to two events: (1) expression of LTP or LTD, and (2) setting of a synapse-specific eligibility trace ('tagging') which allows modulation by protein synthesis-dependent signaling. A major difference is that our findings highlight a role of DA as a key modulator of the eligibility trace. The mechanistic basis for the surprising involvement of protein synthesis in this rapidly induced form of plasticity remains to be investigated.

It has been reported that anisomycin can potentiate JNKs (*Iordanov et al., 1997*). We cannot exclude the possibility that the drug may have affected intracellular signaling cascades that interfere with the plasticity signaling pathway described here.

Our experimental findings uncover a synaptic learning rule that could support a two-stage model of hippocampal memory formation (*Buzsáki, 1989*), in which eligibility traces are laid down during hippocampal theta activity with subsequent postsynaptic burst reactivation during sharp wave-ripples inducing LTP at those synapses.

We considered the activity during navigation on the maze as the animal approaches the reward resembling the STDP priming protocol. Substantial evidence supports a role of NMDAR-dependent STDP in the formation of place fields during navigation (*Mehta, 2015*; *Moore et al., 2021*). It has been postulated that both LTP and LTD are involved in place field formation. This was based on the observation that place fields shift backward with experience (*Mehta et al., 1997*), and a computational model predicted that without LTD place field broadening would occur. Thus LTP is required when entering the place field, and LTD when the animal exits the place field (*Mehta et al., 2000*). This is specific to navigation, as opposed to just walking on a linear track without task, and place field plasticity is predictive of navigational performance (*Moore et al., 2021*).

Exploring possible computational implications of this synaptic learning rule, we first tested it with a neural network in which the most excitable cells are reactivated. The memory allocation hypothesis suggests that learning triggers a temporary increase in neuronal excitability, enabling the linking of individual memories acquired close in time (*Silva et al., 2009*; *Cai et al., 2016*; *Sehgal et al., 2018*). We found that our learning rule selectively strengthens the reactivated synapses, linking together the memory allocation hypothesis with burst reactivation-induced plasticity. Interestingly, it was recently reported that DA released by locus coeruleus cells projecting to dCA1 has a key permissive role in contextual memory linking (*Chowdhury et al., 2021*). Moreover, the rule could also accommodate a temporally discontiguous instructive learning signal or a specific supervisory feedback signal. Thus, it is possible that DA serves as a scalar reward signal which combines with a vectorial representation of the reward identity which triggers the reactivation of a specific subset of neurons. Thus, it was suggested that the direct pathway from the entorhinal cortex could provide an instructive signal to generate accumulation of CA1 place cells at the reward location (*Grienberger and Magee, 2021*). An attractive possibility is that a downstream subset of neurons active during navigation serves as an instructive input onto upstream neurons during reward. This would be consistent with a dual role of DA as reinforcement signal and enhancer of reverse replay (*Ambrose et al., 2016*), establishing a predictive chain of potentiated synapses toward the rewarded outcome reminiscent of the successor representation (*Dayan, 1993*; *Stachenfeld et al., 2017*). It was suggested in a computational study that feedback regulation of synaptic plasticity by bursts in higher hierarchical circuits can coordinate lower-level connections (*Payeur et al., 2021*). Our results reveal a possible biological substrate to support such a mechanism and re-emphasize the importance of bursting activity for synaptic plasticity (*Lisman, 1997*; *Pike et al., 1999*).

The hippocampus receives dopaminergic input from two main sources, the ventral tegmental area (*Scatton et al., 1980*; *Gasbarri et al., 1994*), signaling reward, and the locus coeruleus (*Smith and Greene, 2012*; *Kempadoo et al., 2016*), thought to signal novelty (*Takeuchi et al., 2016*; *Wagatsuma*

et al., 2018), but more recently implicated also in spatial reward learning (*Kaufman et al., 2020*). In a reward location learning task, we found that calcium signals during locomotion were higher in CA1 principal cells that reactivated at reward location. The signal increased in the trial preceding the first reactivation, following which the signal remained higher in all subsequent trials. Moreover, place cells that reactivated showed higher spatial information than non-reactivating place cells in late trials. The overall stability of calcium signals and spatial information is broadly consistent with earlier reports using one-photon imaging in freely moving mice (*Ziv et al., 2013*). Although it was recently reported in head-fixed mice that reactivation increases long-term stability of place cells that have fields distant from the reward after several days (*Grosmark et al., 2021*), we did not see a significant increase in place cell stability after a single reactivation of the cell at the reward location. Irrespectively, the difference in both calcium signal and spatial information in reactivating *vs* non-reactivating cells is suggestive of plasticity related to reactivation events specific to reward location. More work will be required to establish under what conditions the novel burst-induced potentiation mechanism is engaged during hippocampus-dependent learning and memory.

## Methods

**Key resources table**

| Reagent type (species) or resource | Designation | Source or reference | Identifiers | Additional information |
|---|---|---|---|---|
| Strain, strain background (*Mus musculus*) | Wild-type C57BL/6 J, *Mus musculus* | Harlan, Bicester, UK or Central Animal Facility, Physiological Laboratory, Cambridge University | | Age range used for slice preparation: 12–19 days. Females and males were used. |
| Strain, strain background (*Mus musculus*) | Adenylate cyclase double knock-out (AC DKO), C57BL/6 J, *Mus musculus* | Mouse line generated by Hongbing Wang and imported from Michigan State University, MI, US | | Age range used for slice preparation: 12–19 days. Females and males were used. |
| Strain, strain background (*Mus musculus*) | Thy1 – GCaMP6f, C57BL/6 J, *Mus musculus* | Jax Ref: *Dana et al., 2014* | 024276 | Only males were used. |
| Chemical compound, drug | Dopamine hydrochloride | Sigma–Aldrich | H8502 | 100 µM |
| Chemical compound, drug | D-AP5 | Tocris Bioscience | 0106 | 100 µM |
| Chemical compound, drug | Nimodipine | Tocris Bioscience | 0600/100 | 10 µM |
| Chemical compound, drug | PKA inhibitor fragment (6-22) amide | Tocris Bioscience | **1904/1** | 1 µM |
| Chemical compound, drug | Anisomycin | Tocris Bioscience | 1290/10 | 0.5 mM |
| Chemical compound, drug | MK801 | Tocris Bioscience | 0924 | 1 mM |
| Software, algorithm | Igor Pro 6.37 | WaveMetrics | | |
| Software, algorithm | Prism 8.2.0 (435) | Graphpad | | |
| Software, algorithm | Matlab R2021a | MathWorks | | |
| Software, algorithm | CalmAn software (version 1.8.5, Python) | *Giovannucci et al., 2019* | | |

## Mice

Experimental procedures and animal use were performed in accordance with UK Home Office regulations of the UK Animals (Scientific Procedures) Act 1986 and Amendment Regulations 2012, following ethical review by the University of Cambridge Animal Welfare and Ethical Review Body (AWERB). All animal procedures were authorized under Personal and Project licences held by the authors.

Mice were housed on a 12 hr light/dark cycle at 19–23 °C and were provided with food and water *ad libitum*. Experiments were carried out on wildtype C57BL/6 J mice (Harlan, Bicester, UK or Central Animal Facility, Physiological Laboratory, Cambridge University), and adenylate cyclase double knockout (AC DKO) mice, which have the genes for both AC1 and AC8 deleted globally. This mouse line was generated as described previously[25] and was imported from Michigan State University, MI, US. For *in vivo* experiments, seven adult males Thy1 – GCaMP6f transgenic mice

were used (*Dana et al., 2014*) (Jax: 024276). Mice were housed with 2–4 cage-mates in cages with running wheels.

## Electrophysiology

### Slice preparation

Mice of both sexes at postnatal day (P) 12–19 were used in this study. Mice were anesthetized with isoflurane (4% isoflurane in oxygen) and decapitated. The brain was rapidly removed and immersed in ice-cold artificial cerebrospinal fluid (ACSF) containing (in mM): 126 NaCl, 3 KCl, 26.4 $NaH_2CO_3$, 1.25 $NaH_2PO_4$, 2 $MgSO_4$, 2 $CaCl_2$, and 10 glucose (pH 7.2, 270–290 mOsm/L). The ACSF solution was continuously bubbled with carbogen gas (95% $O_2$, 5% $CO_2$). Horizontal slices (350 µm thick) were sectioned with a vibrating microtome (Leica VT 1200 S, Leica Biosystems, Wetzlar, Germany). The slices were then incubated for at least 60 min in ACSF at room temperature in a submerged-style storage chamber before recording. Slices were used for 1–7 hr following sectioning.

### Whole-cell patch clamp recording

For recordings, individual slices were transferred to an immersion-type recording chamber and perfused with ACSF (2 ml/min) at 24–26 °C. Neurons were visualized and selected using infrared differential interference contrast (DIC) microscopy using a 40 X water-immersion objective. The hippo-campal subfields were visually identified and whole-cell patch-clamp recordings were performed on CA1 pyramidal neurons. For stimulation of Schaffer collaterals, monopolar stimulation electrodes were placed in stratum radiatum. Test and control pathway electrodes were placed at the same distance (>100 µm) from and either side of the recorded neuron. Patch pipettes (pipette resistance 4–7 MΩ) were pulled from borosilicate glass capillaries (0.68 mm inner diameter, 1.2 mm outer diameter) using a P-97 Flaming/Brown micropipette puller (Sutter Instruments Co., Novato, California, USA). Pipettes were filled with a solution containing (mM): 110 potassium gluconate, 4 NaCl, 40 HEPES, 2 ATP-Mg, 0.3 GTP (pH 7.2–7.3, 270–285 mOsm/L). The liquid junction potential was not corrected.

All experiments were performed in current-clamp mode. Cells were accepted for the experiment if their resting membrane potential was between −55 and −70 mV. The membrane potential was held at −70 mV throughout the recording by direct current application via the recording electrode. Before the start of each recording, all cells were tested for regular spiking responses to positive current steps— characteristic of pyramidal neurons.

### Stimulation protocol

Excitatory postsynaptic potentials (EPSPs) were evoked alternately in two input pathways (test and control) by direct current pulses at 0.2 Hz (stimulus duration 50 µs) through metal stimulation elec-trodes. Control pathways were used in all experiments to ensure stability control (not always shown). The stimulation intensity was adjusted (100 µA– 500 µA) to evoke an EPSP with peak amplitude between 3 and 8 mV. After a stable EPSP baseline period of at least 10 min, STDP was induced in the test pathway by repeated pairings of single evoked EPSPs and single postsynaptic action potential elicited with the minimum somatic current pulse (1–1.8 nA, 3ms) via the recording electrode. Pairings were repeated 100 times at 0.2 Hz. Spike-timing intervals (Δt in ms) were measured between the onset of the EPSP and the onset of the action potential.

Alternate stimulation of EPSPs was resumed immediately after the pairing protocol and monitored for at least 40 min. For the burst stimulation protocol, stimulation of EPSPs was not resumed for an additional 10 min, and at the end of that period, six bursts, each of five action potentials at 50 Hz, were elicited with an inter-burst interval of 0.1 Hz by somatic current pulses (1.8 nA, 10 ms) via the recording electrode. In a subset of experiments, only five bursts were applied, which led to potentia-tion of a similar magnitude. Immediately after the bursts, stimulation of EPSPs was resumed and moni-tored for at least 30 min, however, a small subset of recordings were stopped at 28 min. Presynaptic stimulation frequency to evoke EPSPs remained constant throughout the experiment. The unpaired pathway served to verify input-specificity and as a stability control. The burst stimulation protocol is summarized in *Figure 1a* (top).

## Drugs

Drugs were bath-applied to the whole slice through the perfusion system by dilution of concentrated stock solutions (prepared in water or DMSO) in ACSF, or by adding the drugs to the patch pipette solution when it was applied intracellularly to the postsynaptic cell only. If the drug was not water-soluble, vehicle control experiments were carried out. For each set of recordings, interleaved control and drug conditions were carried out and were pseudorandomly chosen. The following drugs were used in this study: 100 µM dopamine hydrochloride (Sigma–Aldrich, Dorset, United Kingdom), 100 µM D-AP5 (Tocris Bioscience, Bristol, United Kingdom), 10 µM nimodipine (Tocris Bioscience), 1 µM PKA inhibitor fragment (6-22) amide (Tocris Bioscience), 0.5 mM anisomycin (stock solution in EtOH; Tocris Bioscience), and 1 mM MK801 (Tocris Bioscience).

## Data acquisition and data analysis of slice recordings

Data were collected using an Axon Multiclamp 700B amplifier (Molecular Devices, Sunnyvale, California, USA) and filtered at 2 kHz. Data were acquired and digitized at 5 kHz using an Instrutech ITC-18 A/D interface board (Instrutech, Port Washington, New York, USA) and custom-made acquisition procedures in Igor Pro (WaveMetrics, Lake Oswego, Oregon, USA).

All experiments were carried out in current clamp ('bridge') mode, and only cells with an initial series resistance between 9 and 16 MΩ were included. Series resistance was compensated for by adjusting the bridge balance, and data were discarded if series resistance changed by more than 30%. Offline analyses were done using custom-made procedures in Igor Pro. EPSP slopes were measured on the rising phase of the EPSP as a linear fit between the time points corresponding to 25–30% and 70–75% of the peak amplitude. For statistical analysis, the mean EPSP slope per minute of the recording was calculated from 12 consecutive sweeps and normalized to the baseline (each data point in source data files is the mean of 12 averaged EPSPs). Normalized EPSP slopes from the last 5 min of the baseline (immediately before pairing) and from the last 5 min of the recording were averaged. The magnitude of plasticity, as an indicator of change in synaptic weights, was defined as the average EPSP slope after pairing expressed as a percentage of the average EPSP slope during baseline.

## Statistical analysis of slice recordings

Statistical comparisons were performed using one-sample two-tailed, paired two-tailed, or unpaired two-tailed Student's t-test, with a significance level of $\alpha=0.05$. Data are presented as mean ± SEM. Significance levels are indicated by *p<0.05, **p<0.01, ***p<0.001. All datasets passed the test for normality using the Shapiro-Wilk test ($\alpha=0.05$).

## Computational modeling

We simulated a set of n=10 output neurons, which each received input from 10 input neurons. When an instructive input was added, output neurons additionally received input from two out of 10 instructive neurons. Each output neuron projected uniquely onto one readout neuron which again projected back to the output neuron in a one-to-one mapping. Each output neuron was modeled as an Integrate-and-Fire neuron and 100 trials were simulated, where the voltage v is described by

$$\tau_v \frac{dv}{dt} = -v + w_{inp \to o}I_{inp} + w_{inst \to o}I_{inst} + w_{read \to o}I_{read} + I_{intrinsic}$$

where $\tau_v = 10ms$ is the membrane time constant, $I_{inp}$ are the spike trains of the input neurons, $I_{inst}$ are the spike trains from the instructive neurons, $I_{read}$ are the spike trains from the read-out neurons, $w_{inp \to o}$ are the weights from the input to the output neurons, $w_{inst \to o} = 1/N$ are the weights from the instructive neurons to the output neurons, and $w_{read \to o} = 1$ are the weights from the readout to the output neurons. In addition, each neuron was receiving an intrinsic current $I_{intrinsic} = \alpha_{intrinsic}\eta$, where $\eta$ is simply white noise drawn from a uniform distribution between 0 and 1 and $\alpha_{intrinsic}$ is the magnitude of the current ($\alpha_{intrinsic}$ was applied to all neurons in *Figure 6B*, while in the other configurations no intrinsic current was applied). When the voltage crosses the firing threshold = 0.4, the neuron is reset to the resting potential v=0. Each read-out neuron was also modeled as an Integrate-and-Fire neuron

$$\tau_v \frac{dv_{read}}{dt} = -v_{read} + w_{o \to read}I_o + I$$

where $I_o$ are the spike-trains of the output neurons, $w_{o\rightarrow read} = 1/N$ are the weights from the output to the readout neurons, and $I$ are the spike trains from the supervised neurons. Similarly, if the voltage crossed the firing threshold = 0.4, the neurons emit a spike and the voltage is reset to 0. $w_{inp\rightarrow o}$ were plastic under the following rule. Every time the input neurons are firing (at $t^{pre}$), they are leaving a trace $x^{pre}$, $\tau_{STDP}\frac{dx^{pre}}{dt} = -x^{pre} + \delta\left(t - t^{pre}\right)$, where $\tau_{STDP} = 10ms$ is the trace time constant. Similarly, every time the output neuron spikes (at $t^{post}$), it is leaving a trace $x^{post}$, $\tau_{STDP}\frac{dx^{post}}{dt} = -x^{post} + \delta\left(t - t^{post}\right)$.

The eligibility trace e is described as

$$\tau_e \frac{de}{dt} = -e + x^{pre}x^{post}$$

where $\tau_e = 10$ min is the eligibility time constant. This value was based on the experimental protocol, where burst reactivation was applied 10 min after priming. The $w_{inp\rightarrow o}$ are potentiated if there is an eligibility trace together with dopamine and a postsynaptic burst:

$$w_{inp\rightarrow o} = \alpha_{LTP}deb$$

where $\alpha_{LTP} = 0.0002$ is the learning rate, d=0 if there is no dopamine, and d=1 if there is dopamine.

The postsynaptic burst b is detected as follows. We first computed a trace $x^{burst}$ as

$$\tau_{burst}\frac{dx^{burst}}{dt} = -x^{burst} + \delta\left(t - t^{post}\right),$$

where $\tau_{burst} = 5ms$ is the burst time constant. We set b=1 if $x^{burst} > burst_{threshold}$, to 0 otherwise, with $burst_{threshold}$ = 1.1. If there was no dopamine nor bursts during the whole trial, then the updated rule resulted in a depression

$$w_{inp\rightarrow o} = \alpha_{LTD}e$$

where $\alpha_{LTD} = \alpha_{LTP}$ /500 is the learning rate for the depression. Weights are bound to stay positive. The weights $w_{inp\rightarrow o}$ are initialized to 1 /N. We simulated the network for 2 s in *Figure 6* and 2.5 s in *Figure 7*. The input neurons were firing Poisson statistics at 20 Hz for the first 500 ms of the trial (pairing phase). Dopamine was present during the last 500 ms of the trials in *Figure 6* and the last 1 s in *Figure 7*. The network was simulated for 100 trials (error bars are standard deviations).

In *Figure 6A*, a standard reinforcement learning rule was used: $w_{inp\rightarrow o} = w_{inp\rightarrow o} + \alpha_{RL}de$, where $\alpha_{RL} = \alpha_{LTP}$ /50. In *Figure 6B*, a burst dependent rule was used. Output neurons 3 and 6 had an increased excitability. We modeled that by decreasing the firing threshold to 0.25. All neurons also received an intrinsic current with $\alpha_{intrinsic} = 0.06$ during the last 500 ms of each trial. In *Figure 7A*, the instructive neuron 3 fired Poisson statistics at 500 Hz from time 1.5–2 s and the instructive neuron 6 fired Poisson statistics at 500 Hz from time 2–2.5 s. In *Figure 7B* ii/iii, the read-out neuron 3 received additional Poisson inputs at 60 Hz from time 1.5–2 s and the read-out neuron 6 from time 2–2.5 s. For *Figure 7B* iv, however, the read-out neurons 3 and 6 received their additional Poisson inputs at the same time, from 1.5–2.5 s. In *Figure 7B*, we added lateral inhibition, where each read-out neuron received an inhibitory filtered version (with a time constant of 50 ms) of the spike trains of other read-out neurons, with weights of –1 /N (without self-connection).

## Behavior
### Surgery
Mice underwent two surgeries: the first one to implant a GRIN lens directly above the cells of interest, and the other to fix an aluminum baseplate above the GRIN lens for later attachment of the miniature microscope. The procedures followed the protocol described before by *Resendez et al., 2016*. Surgeries were carried out following minimal standard for aseptic surgery. Analgesic (Meloxicam, 2 mg·kg-1 intraperitoneal) was administered 30 min prior to surgery. Mice were anesthetized with isoflurane (5% induction, 1–2% maintenance, Abbott Ltd, Maidenhead, UK) mixed with oxygen as carrier gas (flow rate 1.0–2.0 L·min⁻¹) and placed in a stereotaxic frame (David Kopf Instruments, Tujunga, CA, USA). The skull was exposed by making skin incision and Bregma and Lambda were aligned horizontally. A 1.5–2 mm-wide craniotomy was drilled above the implantation site. The brain tissue above the implantation site was aspirated. Buffered ACSF was applied throughout the aspiration to prevent desiccation of the tissue. A GRIN lens (1 mm diameter, 4.3 mm length, 0.4 pitch, 0.50

numerical aperture, Grintech) was stereotaxically lowered at coordinates –1.75 AP, 1.75 ML, 1.35–1.40 DV (in mm from Bregma) and fixed to the skull surface with ultraviolet-light curable glue (Loctite 4305) and further fixed with dental adhesive (Metabond, Sun Medical) and dental acrylic cement (Simplex Rapid, Kemdent). A metal head bar was attached to the cranium using dental acrylic cement for head-fixing the animal during the microscope mounting.

If the GCaMP6f expression was visible in the implanted mouse, 4 weeks later the animals were anesthetized for the purpose of attaching a baseplate for the microscope above the top of the GRIN lens. The baseplate was cemented into place and the miniscope was unlocked and detached from the baseplate.

## Behavioral learning task

The mice performed a rewarded spatial navigation task on a round-shaped maze (cheeseboard maze; *Dupret et al., 2010*). The 120 cm diameter cheeseboard had a total of 177 evenly spaced wells. The rewarded wells were baited with ~100 µL of condensed milk mixed 1:1 with water.

For the first three days, the mice foraged for rewards baited in randomly selected wells. A different, random set of wells was baited in each trial. Next, we performed a spatial learning task. The mice had to learn two locations with baited wells. The baited wells had fixed locations that were at least 40 cm apart, chosen pseudo randomly for each mouse. Mice started the trial in one of the three locations on the maze: south, east or west. The maze was rotated and wiped with a disinfectant (Dettol) in-between the trials to discourage the use of intra-maze cues. Landmarks of black and white cues were installed on the walls surrounding the maze. The trials were terminated once the mice ate both rewards or after 300 s. Each learning day consisted of 8 trials with 2–4-minute-long breaks between the trials. To minimize the effects of the novel environment and task structure, we analyzed the neural activity on the first learning day after one of the previous reward locations was moved. After the 5-day-long learning, the memory retention was tested on the next day in a 4 to 5-minute-long unbaited trial. Following the learning and testing of the memory for the first set of locations, we translocated one of the reward locations. The new location was a pseudo-randomly chosen to be at least 40 cm away from the current and previous reward locations. The learning of the new set of locations was performed over two days and tested in an unbaited trial as described above.

The trials were recorded with an overhead webcam video camera. The video was recorded at 24 Hz frame rate. The mice body location was tracked with DeepLabCut software (*Mathis et al., 2018*), a custom-written software was written to map the mouse coordinates to relative location on the maze. The extracted tracks were smoothed with a Gaussian kernel. Periods of running were identified when velocity of the mouse smoothed with a moving average 0.5 s window exceeded 4 cm/s. Immobility was defined as periods of not running that exceeded a duration of 0.5 s.

## Calcium imaging

CaImAn software (version 1.8.5, Python) was used to motion-correct any movements between the calcium imaging frames, identify the cells and extract their fluorescence signal from the video recordings (*Giovannucci et al., 2019*). The method for the cell and signal detection is based on constrained non-negative matrix factorization (CNMF-E; *Pnevmatikakis et al., 2016*). CaImAn extracted background-subtracted calcium fluorescence values and the deconvolved the signal. The deconvolved signal can be interpreted as a scaled probability of a neuron being active. The calcium imaging videos recorded in the same-day trials were concatenated and motion-corrected to a common template frame. Signal extraction and further processing were performed on the resulting long video, allowing to detect the cells and signal present across the trials. To improve the computational performance, the video frames were cropped to a rectangle containing the regions of interest, and the video width and height were downsampled by a factor of 2.

The identified putative cells were automatically filtered using CaImAn. The results were visually inspected and the filtering parameters adjusted to exclude non-cell like shapes and traces from the filtered components. The criteria used for the filtering included a threshold for signal to noise ratio of the trace, the minimum and maximum size of the component's spatial footprint, threshold for consistency of the spatial footprint at different times of the component's activation, and a threshold for component's resemblance to a neuronal soma as evaluated by a convolutional neural network provided with CaImAn software.

The deconvolved trace was time binned, averaging the values in 200 ms bins. A calcium event was detected whenever the cell's deconvolved signal crossed 20% of its day-maximum value. A cell was classified as active during locomotion if it had at least one calcium event. A cell was classified as reactivated if it had at least one calcium event during immobility period and it was active during preceding locomotion. If during the immobility period mice were located within 6 cm of the reward, the reactivation was classified as reactivation at reward, otherwise as reactivation at non-rewarded location. Activity peaks were quantified by their area-under-curve (AUC) which was calculated as a convolution of the preprocessed calcium signal with a 2-s-long flat kernel. The preprocessing of calcium signal subtracted the cell's median value and truncated the values below 0, so that only the above-median calcium signal is integrated in the AUC calculation. We excluded any samples from cells whose maximum value AUC in a given trial did not exceed 0.

## Place cell detection and analysis

To assess how spatial locations modulated activity of a cell, we considered periods of running as described in the 'Behavioral learning task' section and calculated place maps — mean neural activity per spatial bin. The total activity inside 6×6 cm bins was summed from the smoothed deconvolved signal. The mean neural activity in the spatial bin was then calculated as a ratio of the total activity to the total occupancy in the bin after both maps were smoothed across the space using a 2D Gaussian kernel with σ=12 cm. The place map was filtered to include spatial bins with total occupancy that exceeded 1 s (5 time bins, thresholded on unsmoothed total occupancy).

Spatial information of a cell's activity was calculated using the place map values. Spatial information (*Markus et al., 1994*) was defined as:

$$SpatialInformation = \sum_{i=1}^{N} p_i \frac{\lambda_i}{\bar{\lambda}} log_2 \left( \frac{\lambda_i}{\bar{\lambda}} \right)$$

where $\bar{\lambda}$ represents the mean value of the neural signal, $p_i$ represents probability of the occupancy of the i-th bin, and $\lambda_i$ represents its mean neural activity. Dividing by $\bar{\lambda}$ ensures the metric is independent of the cell's average activity. The units of spatial information calculated on calcium fluorescence can be reported as bits per action potential (*Climer and Dombeck, 2021*). However, because the actual action potentials were not measured, we report them as arbitrary units.

Spatial information was compared to the value expected by chance. The chance level was calculated by circularly shifting the activity with regards to the actual location. For each cell, the activity was circularly shifted within the trial by a time offset chosen randomly (minimum offset of 10 s). If the cell's spatial information exceeded 95% values calculated on 1000 random shifts of its activity, it was defined as a place cell.

A limited number of neuronal responses sampled per spatial bin can lead to an upward bias in the estimated spatial information (*Treves and Panzeri, 1995*). To correct this bias, we subtracted its estimated value from the estimated spatial information. The bias was estimated as the mean spatial information from the time-shifting procedure used for place cell detection. This procedure does not require binning of the neuronal responses from the calcium imaging as required by analytical estimation (*Panzeri et al., 2007*), and has been used previously to estimate mutual information bias (*Akrami et al., 2018*).

## Statistical testing of *in vivo* results

To compare the activity peaks between the reactivated and non-reactivated cells throughout the day, we used permutation tests for repeated measures ANOVA (*Figure 8E*, *Figure 8—figure supplement 1A,B*, *Figure 8—figure supplement 2*). The ANOVA modeled fixed effects of trial ordinal and reactivation and random effects of within animal-session factors. Significance level was set to $\alpha$=0.05. To compare the effect of reactivation in each trial on cells grouped by the trial of their first activation, we used permutation t-tests (*Figure 8F*, *Figure 8—figure supplement 1C*). Multiple comparisons were corrected with Benjamini-Hochberg method with the type I error rate set to 0.05. The permutations were restricted to within animal-session permutations. Both ANOVA and t-test statistics were computed based on and 10,000 permutations using 'permuco' R package. Statistical analysis was performed in R version 3.6.3.

## Acknowledgements

This research was supported by the Biotechnology and Biological Sciences Research Council, U.K. We are grateful for discussions of this project with other members of the Neuronal Oscillations Group.

## Additional information

### Funding

| Funder | Grant reference number | Author |
|---|---|---|
| Biotechnology and Biological Sciences Research Council | BB/N019008/1 | Tanja Fuchsberger |
| Biotechnology and Biological Sciences Research Council | BB/P019560/1 | Tanja Fuchsberger |
| Biotechnology and Biological Sciences Research Council | Studentship | Przemyslaw Jarzebowski |

The funders had no role in study design, data collection and interpretation, or the decision to submit the work for publication.

### Author contributions

Tanja Fuchsberger, Conceptualization, Investigation, Writing – original draft, Writing – review and editing, Designed the experiments, Conducted the experiments and analyzed the data, Wrote the manuscript; Claudia Clopath, Software, Investigation, Writing – original draft, Writing – review and editing, Developed the computational model, Wrote the manuscript; Przemyslaw Jarzebowski, Software, Formal analysis, Investigation, Writing – original draft, Writing – review and editing, Conducted the experiments and analyzed the data, Wrote the manuscript; Zuzanna Brzosko, Investigation, Writing – review and editing, Conducted the experiments and analyzed the data; Hongbing Wang, Resources, Provided transgenic AC DKO mice; Ole Paulsen, Conceptualization, Supervision, Funding acquisition, Writing – original draft, Writing – review and editing, Designed the experiments, Wrote the manuscript

### Author ORCIDs

Tanja Fuchsberger (iD) http://orcid.org/0000-0002-4751-8806
Claudia Clopath (iD) http://orcid.org/0000-0003-4507-8648
Ole Paulsen (iD) http://orcid.org/0000-0002-2258-5455

### Ethics

Experimental procedures and animal use were performed in accordance with UK Home Office regulations of the UK Animals (Scientific Procedures) Act 1986 and Amendment Regulations 2012, following ethical review by the University of Cambridge Animal Welfare and Ethical Review Body (AWERB). All animal procedures were authorized under Personal and Project licences held by the authors.

### Decision letter and Author response

Decision letter https://doi.org/10.7554/eLife.81071.sa1
Author response https://doi.org/10.7554/eLife.81071.sa2

## Additional files

### Supplementary files

• MDAR checklist
• Source code 1. Code for computational model.

## Data availability

Data availability Experimental data and code are available at: Code for computational model and code for *in vivo* analysis (including a link to *in vivo* data) are available at: https://github.com/przemyslawj/dCA1-reactivations copy archived at swh:1:rev:22a4e82293f6c36c6fef8c0f06c3f6c68c4045ad. Data of plasticity experiments and of simulation data from computational model are available at: https://data.mendeley.com/datasets/dx7cdgpcz3/1.

The following datasets were generated:

| Author(s) | Year | Dataset title | Dataset URL | Database and Identifier |
| --- | --- | --- | --- | --- |
| Jarzebowski P | 2022 | dCA1-reactivations | https://github.com/przemyslawj/dCA1-reactivations | Github, github.com/przemyslawj/dCA1-reactivations |
| Fuchsberger T | 2022 | Postsynaptic burst reactivation of hippocampal neurons enables associative plasticity of temporally discontiguous inputs | https://doi.org/10.17632/dx7cdgpcz3.1 | Mendeley Data, 10.17632/dx7cdgpcz3.1 |

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
