## [Editor Report]

This article contains fundamental findings that substantially advance understanding of an important research question, mostly using an appropriate and validated methodology in line with the current state-of-the-art, with good and convincing support for the claims. The message of the article will have a profound and lasting influence on neuroscience.

---

## [Decision Letter]

**Decision letter after peer review:**

Thank you for submitting your article "Postsynaptic burst reactivation of hippocampal neurons enables associative plasticity of temporally discontiguous inputs" for consideration by *eLife*. Your article has been reviewed by 3 peer reviewers, and the evaluation has been overseen by a Reviewing Editor and Laura Colgin as the Senior Editor. The reviewers have opted to remain anonymous.

Essential revisions:

1) Further discuss the relationships between burst and DA timing.

2) Acknowledge the relatively non-physiological concentration of calcium used and discuss its influence on the interpretation of the data.

3) Consider including a number of editorial changes as detailed by Reviewer 2.

4) Improvements in methods and statistical analysis, as suggested by reviewer 2.

5) Strengthen the link between the in vitro STDP data as commented by reviewer 3.

*Reviewer #2 (Recommendations for the authors):*

1. The so-called "Hebbian pairing protocol" (e.g., p 4) is not Hebbian. Donald Hebb famously never mentioned synaptic weakening in his books; he only spoke of synaptic strengthening. Also, his postulate also speaks of a cell A that along with a set of other presynaptic cells elicits the spiking in the postsynaptic cell B and how this leads to the strengthening of the connection between A and B. In other words, A necessarily fires before B. In spike-timing-dependent plasticity, the tLTP window is therefore consistent with what Hebb predicted, whereas the tLTD window is neither in disagreement nor agreement with his postulate. Rather, it is an extension of Hebb's postulate. Therefore, the authors should not call this "Hebbian pairing", because it is not. They could, however, call it "spike pairing", "correlated firing", "acausal spiking", or some such thing.

2. It is unclear what is supposed to drive the postsynaptic activity during reactivation, and also when. During sleep? For example, p8 "During DA, information is allocated to primed synapses without the need of further coincident pre- and postsynaptic activity, but by reactivation of the postsynaptic neuron alone." Please clarify here and elsewhere. Please generally try to elaborate on this.

3. The statistical treatment is at times unclear. For example, in Figure 8, "Permutation t-tests with Benjamini-Hochberg correction", but to what value was the false discovery rate set, and how was this value determined? Figure 8F, Stats for React in 4 done over all bins or just 4-8? If the latter, how was this selected for? Same in Figure 8-S1.

4. Electrophysiology methods are at times unclear. Why were the slice experiments carried out at 24-26 {degree sign}C? Why not a more physiological temperature, which is what most labs do? Maybe the eligibility trace decays faster at more physiological temperatures? This would reduce the biological plausibility of this candidate mechanism. P12-19 is not a very mature age. Why were the experiments not carried out in mature animals? I am concerned that these findings might be particular to juveniles. Page 19, "monopolar stimulation electrodes were placed in stratum radiatum", but how far from the recorded CA1 cells? Stimulation too close to the recorded cell is known to possibly activate neuromodulatory fibres such as DA or ACh, which could affect the outcome. Please clarify. It is not immediately clear how many bursts were used. Figures indicate only one arrow, yet the methods state "5-6 bursts" were used, but then later, it says "by somatic current pulses via the recording electrode (5x 1.8 nA, 10 ms each)", where 5x would indicate five bursts. The authors need to be clear in the figures about the precise number of bursts and how many spikes each burst carried. P20, "digitized at 5 kHz", and so filtered at 2.5kHz as per the Nyquist criterion? Please state. P21, R_s selection is unclear, "Series resistance was monitored (10-15 MΩ)", surely the R_s wasn't always <15MOhm, esp. if it could change by as much as 30%, so I am not sure what this means. What was the pipette resistance measured to?

5. Modelling could be clearer. P 22, "tau_e = 10 min is the eligibility time constant. " Where does this value come from? Please justify. P 21, "(α_intrinsic is taken as 0 unless otherwise stated)", please clarify what this means and why it would sometimes be not zero. P26, "Experimental data and code are available at https://github.com/ …" I cannot evaluate code that is not made available.

6. Why is the priming done with post-pre and not pre-post pairing? Conceptually, this part is entirely unclear to me. (related to Major Point 1) Is this choice of timing explained somewhere and I missed it?

7. I struggle with evaluating Figures6 and 7. I simply do not understand what is going on. For example, F7Aii top, Instructive Neurons, is completely blank. The same thing in Bii, Supervised neurons. Why show empty graphs? Why show things that are not talked about in the figure caption text? Figure 8 is also quite unclear.

8. Control experiments seem to be missing or are perhaps just not consistently shown. Please clarify.

– Control experiments (i.e. control pathway) are clear for Figure 1 but are they missing or just not shown elsewhere?

– In Figure 5, Stability control with just anisomycin application seems to be missing too.

– Anisomycin has been shown to result in "profound suppression of neural activity" in the hippocampus (Sharma, Nargang, Dickson, JN 2012), which can affect STDP pairing. Have the authors compared the effects of anisomycin on AP parameters, possibly with anisomycin wash-in?

– Anisomycin can also potentiate JNKs (Iordanov et al. Mol Cell Biol 1997), which are important in synaptic release (e.g. Natisco et al., Sci Rep 2015, Abrahamsson et al., Neuron 2017). It may therefore be helpful to use an alternative protein synthesis inhibitor to confirm the results.

*Reviewer #3 (Recommendations for the authors):*

It would be informative if the authors vary the timing between priming and dopamine application. In their previous work where they used continuous stimulation in the presence of dopamine (Brzosko et al., 2015), a successful potentiation occurs if dopamine is applied immediately after STDP pairing, whereas with a 10 min gap no change is observed. Is the timing between pairing and dopamine application critical or rather the synaptic stimulation (bursting in this case) in the presence of dopamine the point?

Since MPEP blocks presynaptic LTD, it is surprising to me that the amount of potentiation is comparable whether MPEP is present or not (figure 1F vs figure 2A). Any explanation?

Unlike the parts for electrophysiology, the calcium imaging and the navigation-reward task sections are not provided with ample details.

To have a better comparison, the average percentage of reactivated cells at any pair of locations in the maze needs to be calculated. The same analyses shown in Figure 8 need to be done for the locations without rewards.

Page 25: "… The chance level was calculated by circularly shifting the activity with regards to the actual location." Why was circularly shifted activity with a delay used instead of randomly shuffling the activity with regards to the actual location? By shifting, some information still remains in the activity.

Is the increase in the spatial information of a neuron correlated with the temporal gap between its activity during the mice approach and its reactivation at the reward location? On average, does a shorter time gap correlate with a larger activity peak in the following trials?

The reference for Csicsvari et al., has been repeated twice.

---

## [Author Response]

Essential revisions:1) Further discuss the relationships between burst and DA timing.

We have now added this to the Discussion section (page 12, lines 304).

2) Acknowledge the relatively non-physiological concentration of calcium used and discuss its influence on the interpretation of the data.

We have added a discussion of this point on page 12, lines 309.

3) Consider including a number of editorial changes as detailed by Reviewer 2.

We have addressed all of the points that Reviewer 2 raised, and incorporated most of their suggestions into our manuscript. Additionally, several changes on figures have been made as described in detail below (Figure1, 4, 6 and 7).

4) Improvements in methods and statistical analysis, as suggested by reviewer 2.

We have addressed all issues that were raised concerning the Methods section, including the addition of control experiments showing that postsynaptically applied anisomycin has no effect on baseline stability and action potential properties (see supplementary Figure 5-S1). All issues concerning statistical analysis were addressed. Specifically, results for the reward based navigation task have been updated compared with the previous version. We corrected the code that excluded a small number of samples from the results. Some small changes in p-values were further caused by rerunning randomization-based permutation tests.

5) Strengthen the link between the in vitro STDP data as commented by reviewer 3.

Additional analysis of in vivo data has been carried out (pages 9-11, and Figure 8, Figure 8-S1 and Figure 8-S2) to strengthen the link between in vitro and in vivo data. Furthermore, we have added a section to the Discussion providing evidence for the requirement of STDP for place cell formation during navigation (page 13, lines 345).

Reviewer #2 (Recommendations for the authors):1. The so-called "Hebbian pairing protocol" (e.g., p 4) is not Hebbian. Donald Hebb famously never mentioned synaptic weakening in his books; he only spoke of synaptic strengthening. Also, his postulate also speaks of a cell A that along with a set of other presynaptic cells elicits the spiking in the postsynaptic cell B and how this leads to the strengthening of the connection between A and B. In other words, A necessarily fires before B. In spike-timing-dependent plasticity, the tLTP window is therefore consistent with what Hebb predicted, whereas the tLTD window is neither in disagreement nor agreement with his postulate. Rather, it is an extension of Hebb's postulate. Therefore, the authors should not call this "Hebbian pairing", because it is not. They could, however, call it "spike pairing", "correlated firing", "acausal spiking", or some such thing.

We thank the Reviewer for this correction. We agree and have replaced ‘Hebbian’ with ‘spike pairing’.

2. It is unclear what is supposed to drive the postsynaptic activity during reactivation, and also when. During sleep? For example, p8 "During DA, information is allocated to primed synapses without the need of further coincident pre- and postsynaptic activity, but by reactivation of the postsynaptic neuron alone." Please clarify here and elsewhere. Please generally try to elaborate on this.

Following this statement on p8, "During DA, information is allocated to primed synapses without the need of further coincident pre- and postsynaptic activity, but by reactivation of the postsynaptic neuron alone.", the different possibilities, depending on what drives postsynaptic activity, are discussed (page 8, starting from lines 203 from ‘the broader’). To improve clarity, we have now changed the sentence to

“During DA modulation, information is allocated to primed synapses by reactivation of the postsynaptic neuron, and the broader computational implications of this learning rule depend on the control of postsynaptic neuronal bursting activity.”

Briefly, the two main hypotheses are (1) intrinsic excitability according to the memory allocation hypothesis (Yiu et al., 2014) (2) or presynaptic input from other areas (e.g.entorhinal cortex) that encode additional information. Additionally, in the Introduction we give some background on when reactivation occurs and how it is related to neuronal activity (principal neurons fire action potentials in brief bursts during sharp wave ripples) (please see page 3, line 70-77). This is discussed further in the Discussion section on page 14, line 364.

3. The statistical treatment is at times unclear. For example, in Figure 8, "Permutation t-tests with Benjamini-Hochberg correction", but to what value was the false discovery rate set, and how was this value determined? Figure 8F, Stats for React in 4 done over all bins or just 4-8? If the latter, how was this selected for? Same in Figure 8-S1.

For Benjamini-Hochberg corrections, type I error rate was set to 0.05, and we added this information to Methods section (page 30, line 788).

“Multiple comparisons were corrected with Benjamini-Hochberg method with the type I error rate set to 0.05.”

The statistical differences between cell groups shown in Figure 8F were calculated across all trials (trials 1 to 8). We built two statistical models for the effect of reactivation and trial: one model comparing the calcium activity peaks of the reactivated in trial 1 cells with the nonreactivated cells, and another comparing the reactivated in trial 4 cells with non-reactivated cells. The cells reactivated in trial 4 had significantly higher calcium peaks than the nonreactivated cells in trials 4, 5, 7 and 8. Their calcium peaks were not significantly higher in trials 1 to 3 (trials before reactivation), and trial 6 (trial after reactivation). We now state the following in the Figure 8F legend:

“Cells that reactivated in trial 1 had significantly higher normalized calcium peaks in all trials. Cells reactivated for the first time in trial 4 had significantly higher normalized calcium peaks in trials 4, 5, 7 and 8 but not in trial 6 and trials before the reactivation.”

Because the meaning of asterisks marking the significant differences was confusing the

statistically compared values, we removed the asterisks from Figure 8F and Figure 8–S1C. n.s. mark on Figure 8–S1C refers to non-significant interaction between the reactivation and cell type (place cell vs other cell).

We also corrected the n counts in Figures 8F and 8-S1C. Please note that the counts for these panels differ from the ones in Figure 8D. This is because the samples in Figure 8D are restricted for the reactivated cells to the first trial after the reactivation.

4. Electrophysiology methods are at times unclear. Why were the slice experiments carried out at 24-26 °C? Why not a more physiological temperature, which is what most labs do? Maybe the eligibility trace decays faster at more physiological temperatures? This would reduce the biological plausibility of this candidate mechanism.

In our experience, cells in slice recordings at higher temperatures deteriorate more quickly. This is not a problem for shorter recordings, but for spike-timing dependent plasticity we need stable conditions for at least 1 hour during the whole-cell recordings, and it was not feasible to do these recordings closer to body temperature. Furthermore, one of the goals of this study was to compare the signalling mechanism to the one observed during dopamine dependent plasticity with synaptic stimulation (Brzosko et al., 2015), so it was important to keep the conditions as comparable as possible.

P12-19 is not a very mature age. Why were the experiments not carried out in mature animals? I am concerned that these findings might be particular to juveniles.

We agree that, ideally, the ex vivo and in vivo experiments should be done at the

same age.

There are two main reasons for the choice of a younger age for slice preparation. Firstly,

younger tissue is less affected by the sectioning procedure. Secondly, the plasticity induction protocol relies on negative spike pairing, which is typically not leading to synaptic depression in slices of older animals. Thus, the investigation of the conversion from synaptic depression into potentiation required the use of juvenile tissue. However, our findings in vivo suggest that the priming mechanism is physiologically relevant also in adult animals.

Page 19, "monopolar stimulation electrodes were placed in stratum radiatum", but how far from the recorded CA1 cells? Stimulation too close to the recorded cell is known to possibly activate neuromodulatory fibres such as DA or ACh, which could affect the outcome. Please clarify.

We fully agree with the Reviewer that electrical stimulation can potentially lead to the corelease of neuromodulators. In fact, we previously reported that spike pairing at Δt = -10 ms can lead to synaptic potentiation instead of depression due to co-released dopamine (Brzosko et al., *eLife* 2015). Thus care was taken to always place test- and control pathway electrodes at equal distances from the recorded neuron and we ensured the distance exceeded 100 μm. This has been added to the Methods section (page 24, line 576).

Although we cannot exclude the possibility that activation of neuromodulatory fibers couldaffect the outcome of the experiments, this appears unlikely for the experiments

investigating the effect of dopamine and reactivation, since we did not stimulate the test

pathway during bath-application of dopamine.

It is not immediately clear how many bursts were used. Figures indicate only one arrow, yet the methods state "5-6 bursts" were used, but then later, it says "by somatic current pulses via the recording electrode (5x 1.8 nA, 10 ms each)", where 5x would indicate five bursts. The authors need to be clear in the figures about the precise number of bursts and how many spikes each burst carried.

We apologise for the lack of clarity on this. The Methods text has now been updated

accordingly (page 24, line 599), and we now also state this in the Figure legend of Figure 1a.

“For the burst stimulation protocol, stimulation of EPSPs was not resumed for an additional 10 mins and at the end of that period, six bursts, each of five action potentials at 50 Hz, were elicited with an inter-burst interval of 0.1 Hz, by somatic current pulses (1.8 nA, 10 ms) via the recording electrode. In a subset of experiments only five bursts were applied which led to potentiation of similar magnitude.”

P20, "digitized at 5 kHz", and so filtered at 2.5kHz as per the Nyquist criterion? Please state.

Yes, as now stated in the Methods section, the signal was filtered at 2 kHz and sampled at 5 kHz. (page 25, line 622)

P21, R_s selection is unclear, "Series resistance was monitored (10-15 MΩ)", surely the R_s wasn't always <15MOhm, esp. if it could change by as much as 30%, so I am not sure what this means.

We thank the Reviewer for pointing this out and apologize for a lack of clarity. This has now been rewritten (page 25, line 625).

All experiments were carried out in current clamp (‘bridge’) mode, and only cells with an initial series resistance between 9 and 16 MΩ were included. Series resistance was

compensated for by adjusting the bridge balance, and data was discarded if series resistance changed by more than 30%.

What was the pipette resistance measured to?

The pipette resistance was 4–7 MΩ. (page 24, line 577)

5. Modelling could be clearer. P 22, "tau_e = 10 min is the eligibility time constant. " Where does this value come from? Please justify.

tau_e = 10 min was based on the experimental protocol, where burst reactivation was

applied 10 minutes after priming. This has been added to the Methods section (page 27, line 667).

P 21, "(α_intrinsic is taken as 0 unless otherwise stated)", please clarify what this means and why it would sometimes be not zero.

α_intrinsic was applied to all neurons in Figure 6B, while in the other configurations no

intrinsic current was applied. We have changed the wording of this sentence to make this clearer (page 26, line 655).

P26, "Experimental data and code are available at https://github.com/ …" I cannot evaluate code that is not made available.

Experimental data and code have now been uploaded to github.com and referenced in the manuscript (page 30, line 793).

Code for computational model and code for in vivo analysis (including a link to in vivo data) are available at: https://github.com/przemyslawj/dCA1-reactivations. Data of plasticity experiments and of simulation data from computational model are available at: https://data.mendeley.com/datasets/dx7cdgpcz3/1.

6. Why is the priming done with post-pre and not pre-post pairing? Conceptually, this part is entirely unclear to me. (related to Major Point 1) Is this choice of timing explained somewhere and I missed it?

We chose the negative pairing paradigm in order to be able to clearly distinguish the effect of reactivation-induced plasticity from pairing-induced plasticity. With positive pairing it would be challenging to dissociate one type of potentiation from another and results would be difficult to interpret. We have shown though (Figure 1B MPEP) that t-LTD per se is not needed for reactivation-induced plasticity, thus, we concluded that only coincident activity is needed for priming. We have now stated, including in the abstract, that negative pairing was used.

Furthermore, in a behavioural setting it has been postulated that both LTP and LTD occur during place field formation when an animal navigates through an environment. This was based on the observation that place fields shift backwards with experience (Mehta and McNaughton, PNAS 1997), and a computational model predicted that without LTD, place field broadening would occur (Mehta et al., Neuron 2000). Thus LTP is required when entering the place field, and LTD when the animal exits the place field (Mehta et al., Neuron 2000, Mehta, Hippocampus 2015). These observations support the use of negative spike pairing as a behaviorally-relevant and appropriate model for priming.

This has been added to the Discussion (page 13, line 344).

7. I struggle with evaluating Figures 6 and 7. I simply do not understand what is going on. For example, F7Aii top, Instructive Neurons, is completely blank. The same thing in Bii, Supervised neurons. Why show empty graphs? Why show things that are not talked about in the figure caption text? Figure 8 is also quite unclear.

We thank the Reviewer for pointing out that this was not clear. We have now adapted the layout to aid correct interpretation of these figures (see updated Figures 6 and 7). In certain circumstances, there was no activity during part of the protocol (e.g. no activity of instructive neurons during priming period, but they are active during reactivation period), and it is important to show the absence of activity during the initial period.

8. Control experiments seem to be missing or are perhaps just not consistently shown. Please clarify.– Control experiments (i.e. control pathway) are clear for Figure 1 but are they missing or just not shown elsewhere?

We confirm all experiments were carried out with a control pathway for stability control, but not always shown. In Figure 1, additionally to stability control, it was important to show input specificity of this type of plasticity, thus test and control pathway were shown in all panels. In Figures 2-5, while stability control was carried out, we omitted the control pathway traces for visual clarity. We now state this in the Methods section (page 24, line 590). We have added the control pathway in figures when drugs were used that could affect baseline stability (Anisomycin results, see below).

– In Figure 5, Stability control with just anisomycin application seems to be missing too.

We measured EPSP responses over a 60-minute period (the total time of the experiment) in baseline condition (without pairing or dopamine) and observed that the application of the protein synthesis inhibitor did not induce any change to baseline EPSPs. We agree with the Reviewer that this should be shown, and we have now included these results in our manuscript (supplementary Figure 5-S1a), and results are reported in the text (page 7, line 174).

We confirmed that postsynaptically applied anisomycin did not affect synaptic responses in baseline conditions (95% ± 9.7% vs 100%, t(6) = 0.56, p = 0.59, n = 7) (Figure 5, S1a).

– Anisomycin has been shown to result in "profound suppression of neural activity" in the hippocampus (Sharma, Nargang, Dickson, JN 2012), which can affect STDP pairing. Have the authors compared the effects of anisomycin on AP parameters, possibly with anisomycin wash-in?

We thank the Reviewer for this suggestion. We measured the effect of anisomycin on AP

parameters during pairing (supplementary Figure 5 S1 b and c), and report results in the text (page 7, line 176).

We compared action potential properties during pairing in cells with anisomycin to cells loaded with vehicle controls. Spike amplitude (AM 112 mV ± 3 mV, Vehicle 111 mV ± 3 mV) and spike width (AM 3.3 ms ± 0.2 ms, Vehicle 3.3 ms ± 0.2 ms) showed no significant differences (amplitude t(10) = 0.09050, p = 0.92; width t(10) = 0.1134), p = 0.91, (Figure 5-S1B,C).

– Anisomycin can also potentiate JNKs (Iordanov et al. Mol Cell Biol 1997), which are important in synaptic release (e.g. Natisco et al., Sci Rep 2015, Abrahamsson et al., Neuron 2017). It may therefore be helpful to use an alternative protein synthesis inhibitor to confirm the results.

We thank the Reviewer for raising this important point. Since, in all our experiments,

anisomycin was loaded into the postsynaptic cell through the patch pipette, we would not expect it to affect synaptic release. We confirmed that anisomycin did not affect spike properties during pairing and EPSP responses in baseline condition within the 60-minute period of the experiment (see previous point 8). Nevertheless, we have now added to the discussion that a potential effect on JNKs cannot be excluded (page 13, line 336).

Reviewer #3 (Recommendations for the authors):It would be informative if the authors vary the timing between priming and dopamine application. In their previous work where they used continuous stimulation in the presence of dopamine (Brzosko et al., 2015), a successful potentiation occurs if dopamine is applied immediately after STDP pairing, whereas with a 10 min gap no change is observed. Is the timing between pairing and dopamine application critical or rather the synaptic stimulation (bursting in this case) in the presence of dopamine the point?

Our results showed that AC1/AC8 is required for DA- and burst-induced potentiation to

occur. AC1/AC8 are synergistically activated when the two signals, Gs-coupled dopamine D1/D5 receptor activation and ca2+ influx, occur at the same time (Wayman et al., 1994; Watson et al., 2000; Ferguson and Storm, 2004; Neve et al., 2004). To investigate the precise timing requirements for dopamine-dependent reactivation-induced plasticity further, uncaging of caged DA or optogenetically-induced DA release would be suitable approaches for temporal control of the DA transient. These experiments are beyond the scope of the present study. We have added this to the Discussion section (page 12, line 304).

Since MPEP blocks presynaptic LTD, it is surprising to me that the amount of potentiation is comparable whether MPEP is present or not (figure 1F vs figure 2A). Any explanation?

We have previously shown (Brzosko et al., 2015) that, at least in the case of synaptic

stimulation, DA-induced conversion of t-LTD into t-LTP is mediated through two pathways (de-depression and potentiation). If we assume that these two components occur in parallel and are independent, the extent of the potentiation should not be affected by the block of t- LTD.

The strengthening of synaptic weights occurs within minutes and we found that reactivation Induced plasticity requires protein synthesis. Thus there may be a maximum amount of synaptic receptors that can be recruited in that time. This may be a limiting factor in the amount of potentiation that can be reached, at least within the tested time window. Alternatively, in addition to blocking t-LTD, MPEP may partially block/interfere with signaling independent of DA-burst potentiation.

Unlike the parts for electrophysiology, the calcium imaging and the navigation-reward task sections are not provided with ample details.

We added some missing information to the Methods section in the Calcium imaging

subsection (page 28-29). In the Results section, we now state the statistical results as in the electrophysiology section (pages 9-11).

To have a better comparison, the average percentage of reactivated cells at any pair of locations in the maze needs to be calculated. The same analyses shown in Figure 8 need to be done for the locations without rewards.

The Reviewer is right to suggest a control analysis for the effect of immobility at

non-rewarded locations. Mice stopped a median of 4 times at non-reward locations per trial. The number of stops at different locations does not allow us to compare all possible non-reward location pairs. Instead, we investigated if the reactivation at non-rewarded locations affected the following calcium activity peaks.

We now show % of cells reactivated during stops at non-reward locations in Figure 8B. We did not find an effect of reactivation at non-rewarded locations on calcium activity peaks in the trial following the reactivation (Figure 8D).

Page 25: "… The chance level was calculated by circularly shifting the activity with regards to the actual location." Why was circularly shifted activity with a delay used instead of randomly shuffling the activity with regards to the actual location? By shifting, some information still remains in the activity.

The circular shifts of the activity were performed with a randomly drawn delay. Shifting the activity by different delays for each trial with regards to the location data results in

inconsistent information about the location between the trials, as a result, removing spatial information from the calculated place map. Circularly shifting the activity has become a commonly used method in recent years, for example, see Wills et al., 2010, Meshulam et al., 2017, Grosmark et al., 2021. The main advantage over random shuffles is that it preserves temporal dynamics of neuronal activity, making the generated calcium traces more realistic.

References:

Grosmark, A.D. et al. 2020. Offline Memory Reactivation Promotes the Consolidation Of Spatially Unbiased Long-Term Cognitive Maps’. bioRxiv [Preprint]. Available at: https://doi.org/10.1101/2020.08.20.259879.

Meshulam L, Gauthier JL, Brody CD, Tank DW, Bialek W. 2017. Collective Behavior of Place and Non-place Neurons in the Hippocampal Network. Neuron 96:1178-1191.e4. doi:10.1016/j.neuron.2017.10.027

Wills TJ, Cacucci F, Burgess N, O’Keefe J. 2010. Development of the Hippocampal Cognitive Map in Preweanling Rats. Science 328:1573–1576. doi:10.1126/science.1188224

Is the increase in the spatial information of a neuron correlated with the temporal gap between its activity during the mice approach and its reactivation at the reward location? On average, does a shorter time gap correlate with a larger activity peak in the following trials?

We performed additional analysis to answer these questions. The time elapsed from the last activation of the neuron before the mouse arrived at the reward to the time of neuron’s first reactivation did not correlate with a larger activity peak change in the trial following (result now in Figure 8-S1A). Similarly, there was no correlation between the elapsed time and spatial information change (Figure 8-S2C). The discovered synaptic rule leaves a long-duration eligibility trace suggestive a lack of dependence on the time from the initial activity to the reactivation, consistent with these two results.

The reference for Csicsvari et al., has been repeated twice.

We thank the Reviewer for spotting this. The duplicated reference has been removed.